

# Precipitation-temperature scaling: current challenges and proposed methodological strategies

Matthew B. Switanek[1], Jakob Abermann[1], Wolfgang Schöner[1], and Michael L. Anderson[2]

[1]Department of Geography and Regional Science, University of Graz, Graz, Austria
[2]California Department of Water Resources, Sacramento, California

**Correspondence:** Matthew Switanek (matthew.switanek@uni-graz.at)

**Abstract.** Sub-daily to daily extreme precipitation intensities are expected to increase in a warming climate, consistent with the Clausius-Clapeyron (CC) relationship, which predicts a ∼7% increase in atmospheric moisture-holding capacity per °C of warming. Many studies have benchmarked observed extreme precipitation–temperature (P–T) scaling rates against this theoretical value, finding that global averages align closely with CC, while regional and seasonal estimates often diverge
substantially. Significant challenges remain, however, in accurately estimating and interpreting P–T scaling rates, particularly at point scales. In this study, we use observational data from the Upper Colorado River Basin to explore these challenges and propose methodological improvements. Specifically, we compare multiple approaches, including those using raw (non-normalized) and normalized data, to estimate P–T scaling for hourly and daily extreme precipitation. Model performance is assessed using a cross-validation framework. Our results demonstrate that normalizing data, independently for every station
and each calendar month, is essential to account for spatial and temporal climatological variability. Without normalization, estimated scaling rates can be inaccurate and misleading.

## 1 Introduction

Climate change is expected to increase the intensity of extreme precipitation events lasting from sub-daily to daily timescales across many regions of the world (Lenderink and van Meijgaard, 2008; Lenderink et al., 2011; Ban et al., 2015; Tabari, 2020;
Abatzoglou et al., 2022; Ali et al., 2022; Chiappa et al., 2024; Marra et al., 2024). This increase can primarily be attributed to the increased moisture-holding capacity of a warmer atmosphere (Alduchov and Eskridge, 1996; Allen and Ingram, 2002; Lenderink and van Meijgaard, 2010; Huang and Swain, 2022; Gu et al., 2023; Trenberth et al., 2003; Rahat et al., 2024). The Clausius-Clapeyron relationship defines the theoretical rate at which the moisture-holding capacity of the atmosphere scales with temperature. It states that there is approximately a 7% increase in moisture-holding capacity of the atmosphere for every
1.0°C increase in temperature. Extreme precipitation events depend on the available moisture in the air column, and hence, the intensities of those extremes are also expected to increase (Panthou et al., 2014; Westra et al., 2014; Wasko et al., 2015; Myhre et al., 2019; Fowler et al., 2021; Gründemann et al., 2022; Harp and Horton, 2022; Donat et al., 2023).

Over the last couple of decades, a growing body of research has emerged concerning the estimation and application of precipitation-temperature (P-T) scaling rates (Lenderink and van Meijgaard, 2008; Berg et al., 2009; Lenderink and van Meij-



gaard, 2010; Lenderink et al., 2011; Ban et al., 2015; Prein et al., 2017; Fowler et al., 2021; Ali et al., 2022; Dollan et al., 2022; Marra et al., 2024). Many studies have used observations in an effort to better quantify P-T scaling rates (Jones et al., 2010; Lenderink et al., 2011; Utsumi et al., 2011; Ali et al., 2018; Wasko et al., 2018; Ali et al., 2021; Najibi and Steinschneider, 2023), while others have investigated P-T scaling rates using climate model data (Ban et al., 2015; Drobinski et al., 2018; Meresa et al., 2022; Donat et al., 2023; Jong et al., 2023; Martinez-Villalobos and Neelin, 2023; Chiappa et al., 2024; Ester-

mann et al., 2025; Higgins et al., 2025). Past efforts to better quantify and/or estimate P-T scaling rates can be further separated by the choices of temporal and spatial extents. Some research has concentrated on daily extreme precipitation (Utsumi et al., 2011; Ali et al., 2018; Yin et al., 2021), while others have focused on hourly extreme precipitation (Lenderink et al., 2011; Prein et al., 2017; Ali et al., 2021). Likewise, the spatial extent of some prior work has been at the global scale (Ali et al., 2018; Tabari, 2020; Tian et al., 2023), while others have investigated scaling rates at the point or regional scale (Jones et al., 2010;

Drobinski et al., 2018; Najibi et al., 2022; Martinez-Villalobos and Neelin, 2023).

Estimates of P-T scaling rates have primarily been obtained by conditioning extreme precipitation on either 2-meter air temperature (i.e., dry-bulb temperature) (Jones et al., 2010; Utsumi et al., 2011; Panthou et al., 2014; Wasko et al., 2015; Prein et al., 2017; Li et al., 2023; Marra et al., 2024) or 2-meter dew point temperature (Lenderink and van Meijgaard, 2010; Zhang et al., 2017; Wasko et al., 2018; Najibi and Steinschneider, 2023). For the most part, studies which have used dew point

temperature have found greater consistency and more robust relationships than when using air temperature (Lenderink and van Meijgaard, 2010; Lenderink et al., 2011; Ali and Mishra, 2017; Wasko et al., 2018; Barbero et al., 2018). This can be attributed to the fact that dew point temperature also contains information concerning the available moisture in the atmosphere.

In addition to decisions concerning which data to use, prior reseach has also proposed a variety of methods to estimate P-T scaling rates. One of the most widely used approaches is the binning method (Lenderink et al., 2011; Prein et al., 2017; Ali et al.,

2018; Drobinski et al., 2018; Fowler et al., 2021; Gu et al., 2023; Tian et al., 2023; Marra et al., 2024). The binning method has been applied, for example, to investigate how extreme daily precipitation changes as a function of dew point temperature (Ali et al., 2018; Wasko et al., 2018). Often, the binning method is used with data pooled from more than one station in the same region (Utsumi et al., 2011; Drobinski et al., 2018). Pooling data in this manner can leverage an increased sample size in an effort to improve the robustness of the estimates (Molnar et al., 2015; Ali et al., 2021). The binned averages, with or without

pooling, provide estimates of how extreme precipitation scales as a function of the chosen temperature variable. It is important to recognize, however, that climatological differences can be present in the data across both time and space (e.g., California has more extreme daily precipitation in the winter at lower dew point temperatures than it does in the summer). Due to these climatological differences, one runs the risk of inaccurately estimating the apparent scaling rates if data normalization is not applied. Notably, Zhang et al. (2017) proposed a method which uses normalized data computed at the station-level across a

three to four month season. Their method removes many of the climatological differences that are present in the data. However, even climatological differences across different months within a 3-month season can potentially have a large impact on the estimated scaling rates.

Another topic that has recieved significant attention is the so-called "hook" or peak structure which has been shown to be present in scaling rates at higher temperatures (Prein et al., 2017; Wang et al., 2017; Drobinski et al., 2018; Yin et al.,



2018; Fowler et al., 2021; Yin et al., 2021; Gu et al., 2023; Marra et al., 2024). This hook can be described as a shift from positive to negative scaling rates. A number of studies have found the "hook" pattern to be less prevalent when using dew point temperatures instead of air temperatures (Lenderink and van Meijgaard, 2010; Lenderink et al., 2011; Zhang et al., 2017; Ali et al., 2018; Wasko et al., 2018; Fowler et al., 2021). However, some cases still show a "hook" pattern in the scaling rates which have conditioned extreme precipitation on dew point temperature (Lenderink and van Meijgaard, 2010; Lenderink et al., 2011;

Panthou et al., 2014; Tian et al., 2023; Sokol et al., 2024). Different physical mechanisms, such as limited moisture availability and atmospheric dynamics, have been proposed to explain the transition from positive to negative scaling (Berg et al., 2009; Jones et al., 2010; Utsumi et al., 2011; Molnar et al., 2015; Sun and Wang, 2022; Gu et al., 2023). Boessenkool et al. (2017) have shown that sample size can also play a key role in the robustness of the estimates at higher temperatures.

    Our aims in this paper are twofold. First, we use the binning method to illustrate and describe some common challenges or

problems that exist when estimating and interpreting scaling rates. Second, we suggest a methodology to resolve many of those problems which is an extension of the work by Zhang et al. (2017). Lastly, we use three different methods to produce estimates of P-T scaling rates, and use these estimates to generate predictions of extreme hourly and daily precipitation through a cross-validated framework. The skill of these predictions are subsequently evaluated against climatology and against assuming a theoretical Clausius-Clapeyron scaling rate. This allows us to quantify the added value of certain methods over others with

respect to our ability to predict changes to extreme precipitation as a function of changes in dew point temperature.

## 2   Data

Hourly measurements of precipitation and dew point temperature for the Upper Colorado River Basin (UCRB) are obtained from the GH2D-MetNet dataset (Switanek, 2025), which is a quality controlled, global dataset of observed precipitation, temperature, and dew point temperature derived from the Global Historical Climatology Network - hourly (GHCN-hourly)

dataset (Smith et al., 2011). Hourly data is used beginning at 00:00, January 1, 1951 and ending at 23:00, December 31, 2023. The dataset is relatively sparse until around the year 1999, when the density of the stations increases. The spatial distribution of these stations can be observed in Fig. 1a.

    Daily measurements of precipitation are taken from the Global Historical Climatology Network - daily (GHCN-daily) dataset (Menne et al., 2012). Many of these stations do not measure dew point temperature in-situ. Therefore, the ERA5 Reanalysis

dataset (Hersbach et al., 2023) is used along with GHCN-daily to provide dew point temperatures at the GHCN-daily stations. For each GHCN-daily station, the nearest ERA5 grid cell is found, and the corresponding time series of dew point temperatures are used for that station. This procedure is repeated for all of the GHCN-daily stations in the UCRB. We use a common period of record for the hourly and daily data between January 1, 1951 through December 31, 2023. The distribution of the daily stations can be observed in Fig. 1b.





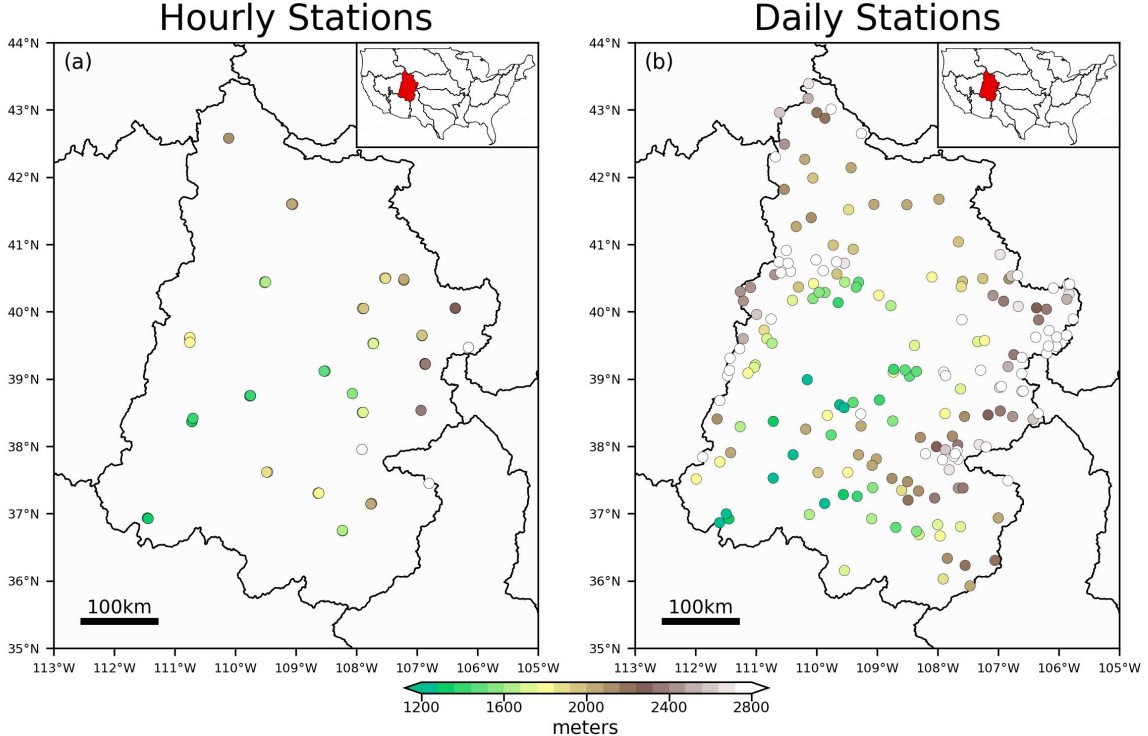

**Figure 1.** The hourly (a) and daily (b) distribution of stations across the Upper Colorado River Basin which are used in this study. The location of the study region is shown, along with other large-scale hydrological basins in the contiguous United States, in the inset subplots in the upper right. The color of the stations corresponds to the station elevation.

## 2.1 Evaluation Metrics Used for Validation

We evaluate model performance across a range of different cases using the root mean squared error skill score ($\mathbf{SS}_{RMSE}$). The skill score, $\mathbf{SS}_{RMSE}$, is a function of the model and reference RMSE errors ($\mathbf{RMSE}_{MOD}$ and $\mathbf{RMSE}_{REF}$, respectively). The $\mathbf{RMSE}_{MOD}$ is defined as,

$$\mathbf{RMSE}_{MOD} = \sqrt{\frac{1}{n}\sum_{i=1}^{n}(y_{mod,i} - y_{obs,i})^2} \ . \tag{1}$$

where $y_{obs}$ and $y_{mod}$ are the observed and the modeled precipitation values, respectively. Likewise, $\mathbf{RMSE}_{REF}$, reflects the error associated with a reference or baseline model.

$$\mathbf{RMSE}_{REF} = \sqrt{\frac{1}{n}\sum_{i=1}^{n}(y_{ref,i} - y_{obs,i})^2} \ . \tag{2}$$

where $y_{ref}$ contains the reference precipitation values. We compare against two reference modeled values. First, we compare whether the model predictions are more skillful than climatology (i.e., always assuming 100% of normal, or equal to the





climatological mean). And second, we compare whether the model predictions are more skillful than if we had assumed a theoretical Clausius-Clapeyron (CC) relationship (i.e., using 7% per °C). The skill score can then be obtained as,

$$\mathbf{SS}_{RMSE} = 1 - \frac{\mathbf{RMSE}_{MOD}}{\mathbf{RMSE}_{REF}} \quad . \tag{3}$$

Skill scores of $\mathbf{SS}_{RMSE}$ above zero indicate that the model predictions are more skillful than the reference predictions, while scores below zero indicate that the model is performing worse than the reference.

## 3 Methods

### 3.1 Common Methodological Challenges Pertaining to the Interpretation of Scaling Rates

Ultimately, our goal is to have a methodological approach that can more accurately estimate P-T scaling rates, and to use those estimates to make skillful predictions of extreme precipitation. We want to be able to estimate scaling rates with sufficient enough spatial and temporal resolution in order to say, for example, that a particular region in a given month or season can

expect some specified percentage increase in extreme precipitation, on average, provided that the region experiences +1°C of warming in dew point temperature. The percentage increase per °C is our scaling rate and it will vary as a function of space (e.g., some chosen region or basin) and time (e.g., month).

Without carefully managing the underlying data, one can incorrectly estimate and interpret P-T scaling rates. There are three primary concerns that must be addressed prior to using the data to estimate scaling rates:

1. Using raw, or non-normalized, values of dew point temperatures and precipitation rates across multiple stations and/or months can lead to an inaccurate estimate of a scaling rate due to climatological differences that exist in both space (from station to station) and time (from month to month).

2. Differences in sample sizes.

3. Data at hourly or daily resolutions cannot be assumed to be temporally independent.

In Figure 2, we plot an example of the first of the aforementioned challenges. This challenge relates to the conflation of different climatologies that can arise in space and time when using measured data that has not undergone any normalization. In Fig. 2a, measured values of dew point temperature and precipitation rates from multiple stations, and across multiple months, are plotted together using a binning method. Figure 2a plots all of the pairings of hourly dew point temperature and precipitation rates using all of the stations that fall within the UCRB. The top 1.0% and top 0.1% of precipitation events are shown for dew

point temperature bins, where the bins are iterated for every 1°C with a bin size of 2°C (e.g., centering at 10°C and using values between 9°C and 11°C, then stepping to 11°C). The observed scaling rates for the top 0.1% and 1.0% are plotted as the solid and dashed-dotted red lines, respectively. The red lines are calculated using the average precipitation of the points that fall in the top 0.1% and 1.0% of each bin. Note, we found that using the median precipitation of the points (which is not shown), instead of the average precipitation, did not change the shape of the scaling rate curves. The binning method which uses the



measured data, seen in Fig. 2a, shows a "hook" pattern at higher dew point temperatures, where the scaling rate transitions from positive to negative (Lenderink et al., 2011; Tian et al., 2023; Visser et al., 2021; Yin et al., 2021; Sokol et al., 2024). This "hook" pattern was also found using different bin sizes. The presence of this "hook" complicates the interpretation of the associated scaling rate, especially at these higher dew point temperatures. It is unclear how exactly one should extrapolate new precipitation extremes given new high-valued dew point temperature extremes. The very presence of the "hook" or peak

pattern, however, could simply be the result of different stations and different months, all with different climatologies, being plotted together. We can test if this is the case here by removing climatological differences in the underlying data. To do this, we can compute z-scores of the hourly time series, month by month and station by station. Hourly z-scores of dew point temperature are computed as,

$$\mathbf{DPT}^z_{x,m,t} = \frac{\mathbf{DPT}_{x,m,t} - \overline{\mathbf{DPT}}_{x,m}}{\sigma_{\mathbf{DPT}_{x,m}}} \quad , \tag{4}$$

where $\mathbf{DPT}_{x,m,t}$ is the dew point temperature at station $x$, month $m$, and hour $t$, and $\overline{\mathbf{DPT}}_{x,m}$ and $\sigma_{\mathbf{DPT}_{x,m}}$ are the mean and standard deviation of the dew point temperature time series at station $x$ and month $m$, respectively. The maximum number of data points in the array, $\mathbf{DPT}_{x,m}$, for the month of July is 54,312 ((24 hours) x (31 days) x (73 years)). Similarly, hourly z-scores of precipitation are computed as,

$$\mathbf{P}^z_{x,m,t} = \frac{\mathbf{P}_{x,m,t} - \overline{\mathbf{P}}_{x,m}}{\sigma_{\mathbf{P}_{x,m}}} \quad , \tag{5}$$

where $\mathbf{P}_{x,m,t}$ is the precipitation rate at station $x$, month $m$, and hour $t$, and $\overline{\mathbf{P}}_{x,m}$ and $\sigma_{\mathbf{P}_{x,m}}$ are the mean and standard deviation of the precipitation time series at station $x$ and month $m$, respectively. Figure 2b plots all of the standardized pairings (from Eqs. 4 and 5) of hourly dew point temperature and precipitation rates using all of the stations that fall within the UCRB. The same data is used to produce Figs. 2a and 2b. However, the data in Fig. 2b has undergone a transformation to remove climatological differences in the underlying data. The "hook" pattern from Fig. 2a is not present in Fig. 2b. We can clearly

observe, in this case, that the "hook" structure in Fig. 2a can be attributed to climatological differences of the underlying data in space and time. After removing these variations in climatology, extreme standardized anomalies of precipitation are found to increase across the entire range of standardized anomalies of dew point temperature. Note, the z-scores of precipitation in Fig. 2b are quite large. This is due to fact that Eq. 5 is computed using the entire time series, which contain many zeros. We also restricted the analysis to using only the data where there was positive precipitation, and we found the same result where

precipitation anomalies are found to continue to increase even at the highest dew point temperature anomalies.

        Figure 3 plots the number of points that fall within the top 0.1% of precipitation values for each of the dew point temperature bins in Fig. 2a. These different number of points corresponding to the bars in Fig. 3 are the various sample sizes used to compute the scaling rates via the binning method. One can clearly observe that the number of extreme precipitation events that reside in the bin centered about 16°C is much less than the number of extreme precipitation events that fall within the bins

between -5°C and 0°C. There are roughly 50 times the number of events at -3°C than at 16°C or 17°C. As a result, the binning method is less robustly estimating what is considered "normal" for the top 0.1% of precipitation events for certain dew point temperatures than for others.





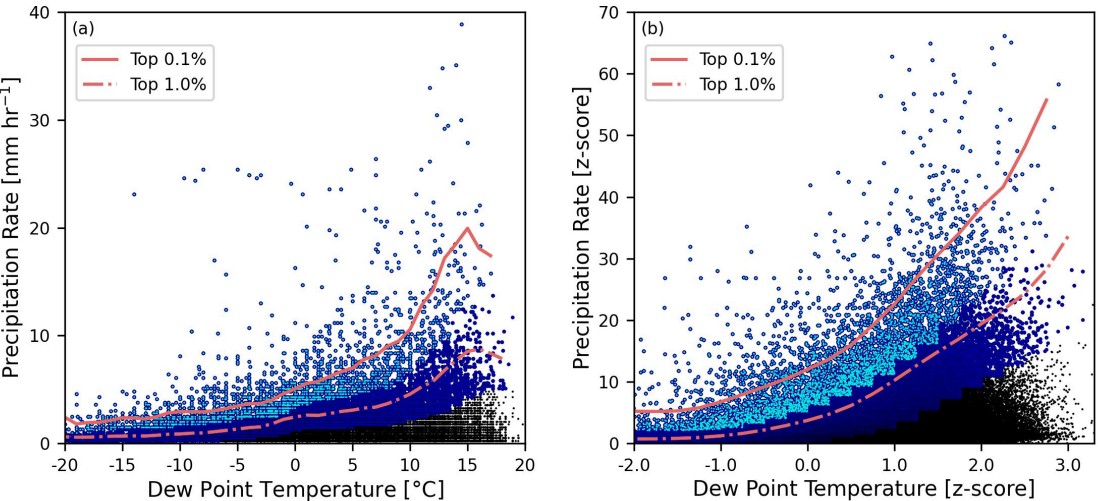

**Figure 2.** (a) All of the pairings of measured hourly precipitation along with the corresponding dew point temperature for the stations in the UCRB are plotted in the background as the black scatter points. The top 0.1% and 1.0% of precipitation rates are plotted as the light blue and dark blue scatter points, respectively. The top 0.1% and 1.0% are for 2-degree bin windows with 1-degree increments. The solid and dashed-dotted red lines are the average precipitation rates for the top 0.1% and 1.0% of values that fall within each of the bins. (b) The same as (a), except that the data has been standardized (i.e., z-scores).

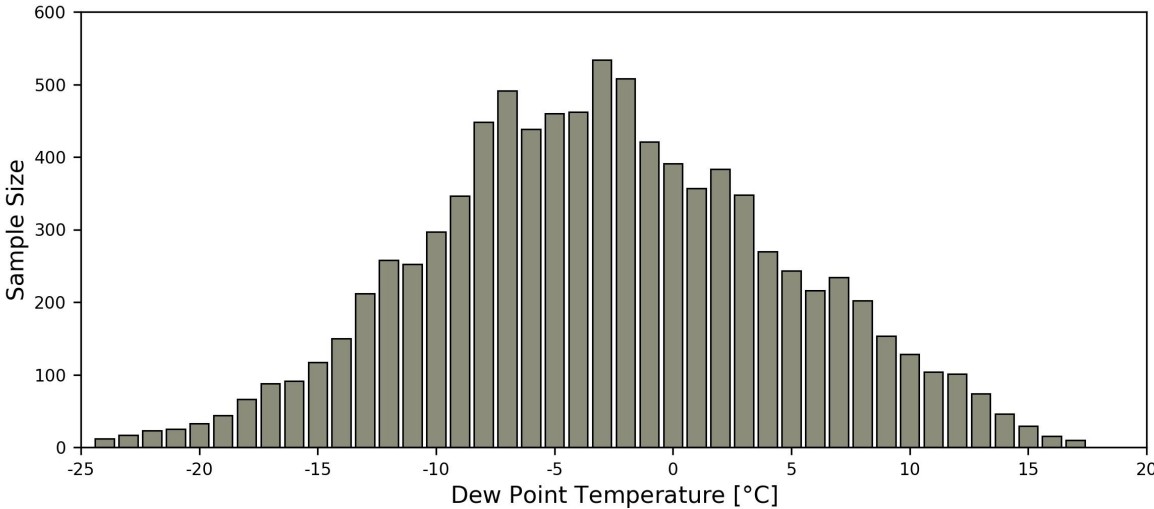

**Figure 3.** The number of samples, or the number of light blue data points, that fall in the top 0.1% for each of the dew point temperature bins from Figure 2a.



Due to the fact that both the dew point temperature and precipitation time series are positively autocorrelated, the effective sample sizes are actually much less than what is suggested in Fig. 3. Focusing on the precipitation time series, Figure 4a

plots the autocorrelation of these time series, station by station, using the hourly data set. For each station, we can take a subset of the time series for which the current hour and the subsequent hour both contained data, and we can compute the lagged-1 autocorrelation. The empirical cumulative distribution function (CDF) of all of the different station autocorrelations from Fig. 4a is plotted as the thicker black line in Fig. 4b. To determine the statistical significance of these autocorrelations, we additionally ran 100 randomized simulations. For each simulation, we randomly generated a shuffled realization of the

precipitation data at each station, drawing upon the empirical distribution of precipitation at that station, and computed the corresponding autocorrelation. The CDFs of these 100 simulations are seen as the thin colored lines in Fig. 4b. The expected autocorrelation is zero through the null hypothesis, and we find that the observed autocorrelation of the hourly precipitation data set to be statistically significantly greater than this null hypothesis (p<0.01). Figures 4c-4d more specifically investigate the conditional probabilities of the extreme precipitation events themselves. Using the hourly data at one station as an example,

we look at what the probability of having an extreme precipitation event in the top 0.1% for some specified hour, $t + 1$, given that the prior hour, $t$, was also an extreme event in the top 0.1%. The spatial distribution of those probabilities are plotted for the hourly data set in Fig. 4c. The CDFs of the probabilities from Fig. 4c are plotted as the thick black line in Fig. 4d. Again, the probabilities are statistically significantly greater than what we would expect by chance (p<0.01), as seen by the thin colored lines. In fact, the average probability that an hourly value of precipitation at a particular station would be in the

top 0.1%, given that the previous hourly precipitation at that same station was also in the top 0.1%, is greater than 100 times more likely than if the data were temporally independent (averge probability of the black line in Figure 4d, which is 0.14, versus the event likelihood, which is 0.001). Figures 4e-4h show the same as Figs. 4a-4d, except now using the daily dataset. Note, that the colorbar ranges are different between Figs. 4e and 4a, and similarly between Figs. 4g and 4c. This indicates, as we would expect, that there is a higher degree of autocorrelation in the hourly temporal resolution than at the daily resolution.

That said, the correlations in Fig. 4e and the probabilities in Fig. 4g, which use the daily data, are also found to be statistically significantly greater than the null hypothesis (p<0.01). For the daily data, it is more than 50 times as likely, than by chance alone, that a daily event at a particular station will be extreme given the prior day was also extreme. Adjacent hourly and daily values of precipitation, and dew point temperature for that matter, cannot be considered statistically independent. As a result, if one were to apply an approach such as the binning method, one cannot include all of the hourly or daily data points and treat

them as statistically independent events.

Still another issue that requires consideration is the collocation in time of dew point temperatures at a given hour or day along with the maximum hourly or daily precipitation rates. Ideally, our goal is to devise methods of finding "historical" scaling rates where we can then predict expected changes in extreme precipitation given changes in our dew point temperatures. To that end, we can think of our maximum hourly precpitation as our dependent variable, and it depends on, or is conditioned on changes

in dew point temperature. Given some "future" projected distribution of hourly dew point temperatures for a given month at a given station, it is not clear ahead of time at which hourly dew point temperature we will observe the maximum hourly precipitation rate. Figure 5a plots three empirical CDFs of hourly dew point temperatures for one randomly selected station for





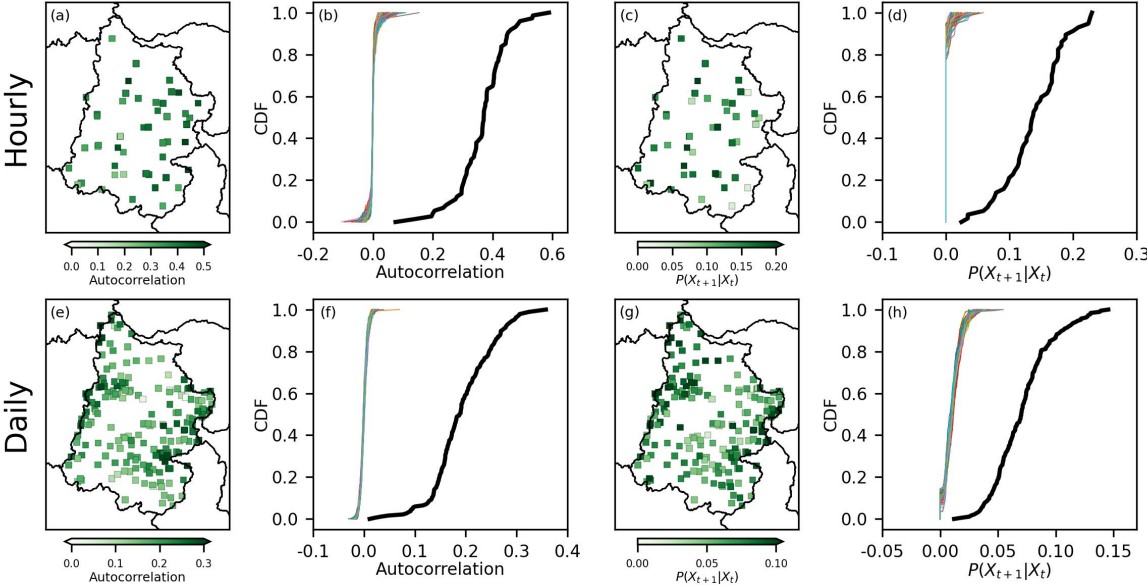

**Figure 4.** The top and bottom rows of subplots corresponds to hourly and daily data, respectively. (a) The lagged-1 autocorrelations of the hourly precipitation time series data are plotted for the stations in the UCRB. (b) The thick black line is the empirical cumulative distribution function (CDF) of all of the station lagged-1 autocorrelations from (a). The empircal CDFs of 100 randomly resampled realizations are the thin colored lines in (b). (c) The probability, at each station, of having an hourly precipitation rate fall in the top 0.1%, given that the previous hour was also in the top 0.1%. (d) The same as (b), except now computing the probabilities, instead of autocorrelations, given the randomized realizations. (e)-(h) These are the same as (a)-(d), except using daily precipitation data.

the month of July. The curves are for the same month, but using three different years. One can clearly observe in Fig. 5a that the distribution of dew point temperature from July 1996 is substantially warmer than the distribution from July 2020. However,

the dew point temperature at which the maximum hourly precipitation occured is lower in the year 1996 than 2020. So, even though the mean of the distribution for the year 1996 is more than 5°C warmer than the mean of the distribution from 2020, the maximum hourly precipitation occured at a lower dew point temperature in 1996 than in 2020. One can also compare the dew point temperature distribution from 1996 to the year 2001. In this case, the dew point temperature distributions are nearly identical, but the dew point temperature at which the most extreme hourly precipitation occured is quite different. To restate the

problem, we cannot know ahead of time at which hourly dew point temperature the most extreme precipitation rate will occur. More often than not, the most extreme precipitation will occur in the upper end, or right-tail, of the dew point temperature distribution. This is not always the case, however, and there are even cases where the most extreme precipitation rate for the month occurs during the coldest recorded hourly dew point temperature for that month. Provided that we cannot know ahead of time at which value of our predictor will correspond to the most extreme precipitation, the predictor can more easily be

thought of as the entire distribution itself and whether or not the mean of that distribution has shifted to the left or right. In fact, we find that using the mean monthly dew point temperature even has the potential to improve the accuracy of the forecasts.





Consider that we find for each station our maximum hourly precipitation values for each month in the time series. We also find, for each station, the dew point temperature at the hour where the maximum hourly precipitation for that month occured, in addition to the average monthly dew point temperatures for each month at each station. This gives us three multidimensional arrays: 1) the maximum hourly precipitation for each month and station, 2) the corresponding, or collocated, hourly dew point temperature corresponding with the time of maximum hourly precipitation from the first array, 3) the average monthly dew point temperature for each month and station. These three arrays each have a size of (70,12,73), where 70 is the number of stations, 12 is the number of months in the calendar year, and 73 is the number of years in the data record. Using Eq. 4, we can compute standardized values of the second and third arrays containing different dew point temperatures, where the $t$, from Eq. 4, now corresponds to the year. Similarly, we can use Eq. 5 to compute standardized values of the first array containing precipitation amounts, where again the $t$, from Eq. 5, corresponds to the year. In Fig. 5b, the scatter points, along with the 2-d histograms, between the first and the second standardized arrays, are plotted for the month of July. We can compare Fig. 5b to Fig. 5c which uses the first and third standardized arrays also for the month of July. There is a stronger statistical relationship between the monthly average dew point temperature and maximum hourly precipitation than the collocated hourly dew point temperature. More generally, we find that during the summer months there is a stronger statistical relationship between the average monthly dew point temperature and the maximum hourly precipitation than when using the collocated hourly dew point temperatures. While in the winter, we find the correlation strength to be of a similar magnitude. It is for these reasons outlined here, that we have chosen to use the average monthly dew point temperatures as our predictor of maximum hourly and maximum daily precipitation rates.

## 3.2 Using Normalized Data

In the previous section, we documented three common challenges or problems that can arise when estimating and interpretating P-T scaling rates. However, all three of these problems can largely be circumvented by implementing the following steps:

1. Using the hourly dataset, find the maximum hourly precipitation rate for each month at each station. Similarly, using the daily dataset, find the maximum daily precipitation rate for each month at each station.

2. Compute monthly averages of dew point temperature for each month at each station.

3. Normalize the data from the prior two steps by computing anomalies of the precipitation and dew point temperature time series.

The normalized dew point temperature anomalies can now be computed as,

$$\mathbf{DPT}^*_{x,m,t} = \mathbf{DPT}_{x,m,t} - \overline{\mathbf{DPT}}_{x,m} \quad , \tag{6}$$

where $\mathbf{DPT}_{x,m,t}$ is average dew point temperature at station $x$, month $m$, and year $t$, and $\overline{\mathbf{DPT}}_{x,m}$ is the mean dew point temperature over the calibration time period at station $x$ and month $m$. Similarly, the anomalies of precipitation can be computed




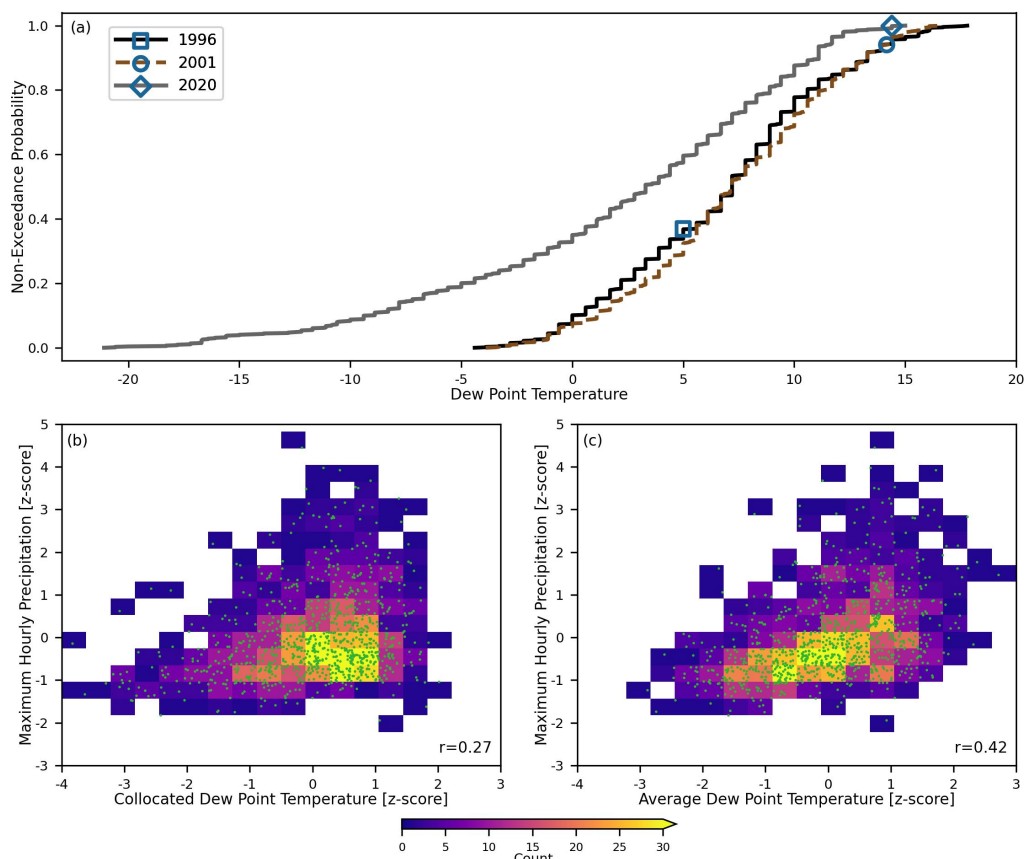

**Figure 5.** (a) The distributions of dew point temperature are plotted for the same station, but for three different Julys. The black line, brown dashed line, and gray line are the distributions for the month of July in the years 1996, 2001, and 2020, respectively. The dew point temperature values are shown on the x-axis, and the y-axis plots the non-exceedance probabilities. The dew point temperature corresponding to the hour at which the maximum hourly precipitation rate occured is enclosed by the hollow blue square, the circle, and blue diamond, respectively. (b) Shows a 2-dimensional histogram between the standardized values of the collocated dew point temperature and maximum hourly precipitation for the month of July (the green dots are the values used to create the histogram). (c) Shows a 2-D histogram between the standardized values of the average monthly dew point temperatures and maximum hourly precipitation for the month of July.

as,

$$\mathbf{P}^*_{x,m,t} = \frac{\mathbf{P}_{x,m,t}}{\overline{\mathbf{P}}_{x,m}} \cdot 100 \quad , \tag{7}$$

where $\mathbf{P}_{x,m,t}$ is either the maximum hourly or maximum daily precipitation rate at station $x$, month $m$, and year $t$, and $\overline{\mathbf{P}}_{x,m}$ is

245  the mean of the respective precipitation (i.e., hourly maximum, daily maximum) time series over the calibration time period at station $x$ and month $m$. By normalizing the data with Eqs. 6 and 7, we have effectively removed the three common challenges or problems that we discussed above. First, we compute anomalies of the data which removes climatological differences between





stations in both space and time. Second, we always use the same sample sizes (e.g., there are always 744 hours and 31 days in January, with the only exception being for the month of February, where there can be small differences in sample sizes given whether or not there is a leap year). And third, we observe the maximum hourly and maximum daily precipitation rates at the station-month scale to be statistically independent. For the duration of the paper, any reference that we make concerning data anomalies or normalized data will correspond to the values computed from Eqs. 6 and 7.

### 3.3 Different Methodological Approaches to Estimating Scaling Rates

In this paper, we use three different methodological approaches to estimate P-T scaling rates. The first uses a binning method with the raw, measured data which has not undergone any transformation or normalization. The second uses a binning method again, but with the normalized data anomalies from Eqs. 6 and 7. For these first two methods, binned-averages are computed over $2°$ windows, or bin widths, and with increments of $1°$. As we have already prescreened the data for the precipitation extremes (i.e., maximum hourly and maximum daily precipitation), we compute binned-averages over all of the existing data points that fall within each bin. The third method fits an exponential curve to the same normalized data used in the second binning method. The model fit is performed using the optimize function contained in Python's Scipy package. This method minimizes the least-squares over the following function:

$$y = a \cdot e^{xb} \quad , \tag{8}$$

where $x$ contains monthly-averaged dew point temperature anomalies, from the array $\mathbf{DPT}^*$, $a$ is a multiplicative offset, and $b$ is interpreted as our scaling rate.

We begin by implementing the two binning methods across a range of different cases of varying spatial and temporal extents. This is so that we can first show the added value of using the normalized data instead of the measured values. Model estimates of P-T scaling rates are used to generate predictions of maximum daily and maximum hourly precipitation at the station-month level, where modeled values are produced for every month at every station. We use the two binning methods to evaluate the cross-validated model performance of the predictions given different combinations of spatial and temporal extents. The five spatial extents that we use are: 1) using only the data from the station being predicted, 2) using stations within a 50 km radius of the station being predicted, 3) using stations within a 100 km radius of the station being predicted, 4) using stations within a 200 km radius of the station being predicted, and 5) using all of the stations that fall within the entire UCRB. The four temporal extents that we use are: 1) the data solely from the month being predicted, 2) a 3-month window centered about the month being predicted, 3) a 5-month window centered about the month being predicted, 4) using all 12 calendar months. Consider a synthetic example where we would like to produce modeled predictions of maximum daily precipitation for station "1" for the month of July over the last 30 years, 1994-2023. We begin by using only the data at this station from all of the Julys over the calibration period 1951-1993. This example corresponds to using the first spatial and first temporal extents. The maximum number of data points used to construct all of the binned values in this case would be 43 (corresponding to the number of data values in July, at one station, over the 43 years in the calibration period). The binned-averaged precipitation rate in this case is often only relying on only a few observations for each temperature bin. Consider another case where we apply the binning





methods using stations within 100 km of station "1" and over a 3-month window corresponding to June-July-August. In this case, let us assume that there were 30 additional stations that fall within 100 km of station "1". Then, the maximum number of data points used to construct the binned values in this case would be 3999 (corresponds to (43 years) x (31 stations) x (3 months)). The same procedure is performed for all combination of cases using the measured and the normalized data.

In Figure 6, we illustrate the three different methods using all of the 187 daily stations that fall within the UCRB (i.e., spatial extent 5) using a 3-month window (i.e., temporal extent 2). Figure 6a shows the binned estimates between the measured monthly-averaged dew point temperatures ($\mathbf{DPT}$) along with the measured maximum daily precpitation amounts ($\mathbf{P}$) using all of the stations in our domain over the winter months December-February. Figure 6b shows the binned estimates between the normalized monthly-averaged dew point temperatures ($\mathbf{DPT}^*$) and the normalized maximum daily precpitation amounts ($\mathbf{P}^*$) for the same stations and months. Figure 6c shows an exponential fit to the same anomalous data from Fig. 6b. Figures 6d-6f show the same as Figs. 6a-6c, but now using the summer months of June-August. The blue and red lines are examples of the modeled P-T estimates used in this paper to produce predictions for the validation periods for these particular cases.

## 3.4 Evaluating Model Performance

The number of observations of daily maximum precipitation is much greater than the number of hourly maximum observations. Therefore, we begin by evaluating the performance of the two different binning models in their ability to predict out-of-sample daily maximum precipitation. This is done using the various combinations of spatial and temporal extents as we discussed in the previous section. At this point, all model predictions are produced using the calibration period 1951-1993 and validated over the period 1994-2023. For example, for a given spatial and temporal extent, the model bins are computed for one station-month using the data over the calibration period 1951-1993. Then, the binned values obtained from the calibration period, are used with the monthly-averaged dew point temperatures from the period 1994-2023 to produce predictions of maximum daily precipitation over the validation period. Model predictions are produced for each month at each station. The first binning method initially produces predictions as non-normalized values of precipitation (i.e., mm per hr). These modeled values are subsequently normalized using Eq. 7. While individual predictions of daily maximum precipitation are produced, the individual values themselves exhibit large fluctuations due to natural variability. This is due to the fact that we are solely conditioning changes in extreme precipitation on changes in dew point temperature. Given the large variability of the individual values themselves, we evaluate model performance as the average change in extreme precipitation given specified dew point temperature anomalies ranging between -3°C below normal to +3°C above normal. At each dew point temperature anomaly, we compute the mean extreme precipitation anomaly centered about that dew point temperauture anomaly. We use a 2° window and find the modeled and observed extreme precipitation distributions corresponding to dew point temperature anomalies of: -3°C, -2°C, -1°C, +1°C, +2°C, and +3°C. Model performance evaluates how well the predicted mean shifts in the normalized extreme precipitation align with observations over the validation period.

Figure 7 provides an illustration of how the model skill score is computed given the case of using the spatial extent of all of the stations within the UCRB (i.e., spatial extent number 5) and the 3-month temporal extent (i.e., temporal extent number 2). In Fig. 7a, the observed monthly-averaged dew point temperature anomalies are plotted against the observed maximum daily





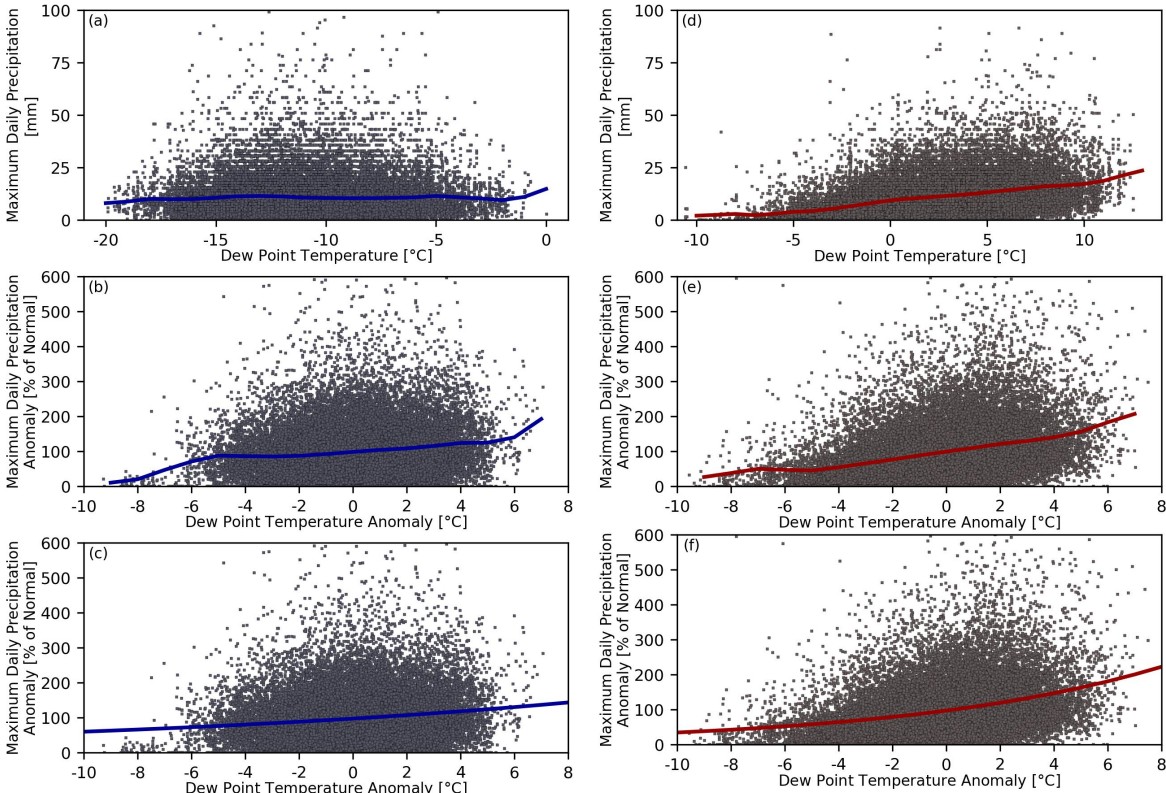

**Figure 6.** The three methods used in this paper are illustrated here. (a) Shows all of non-normalized values of monthly-averaged dew point temperature plotted against maximum daily precipitation rates. All of the daily stations in the UCRB are used, and the data is plotted for the winter months of December-January-February. (b) A binning method is applied again, but now using normalized data. (c) An exponential function is fit to the normalized data. (d)-(f) The same as (a)-(c), except now showing the summer months June-July-August.

precipitation anomalies. The values in Fig. 7a are shown using all of the stations within the UCRB for the month of July over the validation period 1994-2023. There are 4,974 data points in Fig. 7a, which corresponds to a 88% data coverage (88% = 4,974/5,610, where 5,610 is equal to the 30 validation years for the month of July times the 187 stations that reside within the UCRB). The different empirical CDFs of normalized maximum daily precipitation, given the 6 dew point temperature anomalies, are plotted as the curves in Fig. 7b. The mean shifts of the different distributions are plotted as the vertical dotted lines. This gives us 6 observed mean shifts computed over the validation period for the month of July. The same procedure is repeated for all 12 calendar months. Similarly, we can compute the model predicted mean shifts. Then, we can use the modeled and observed mean shifts to compute how skillful the model predictions are (using Eqs. 1-3) with respect to climatology (i.e., 100% of normal), and with respect to a theoretical CC scaling rate of 7% per °C. In Fig. 7c, one can see all of the model predicted means plotted against the observed means. In Fig. 7c, there are 72 scatter points corresponding to the 12 months times the 6 dew point temperature anomalies. The modeled means in Fig. 7c is derived from the binning model which uses





**Figure 7.** (a) Normalized average monthly dew point temperatures, for all daily stations, against normalized maximum daily precipitation for the month of July. This is for the data in the validation period 1994-2023. The different dew point temperature anomalies where model performance is evaluated can be seen as the different colored dashed lines. The orange points located between $0°C$ and $+2°C$ are used to construct the precipitation distribution corresponding to a $+1°C$ dew point temperature anomaly. (b) Plots the distribution of the orange points highlighted in (a) as the thicker orange line. The mean shift of this example distribution is the thicker vertical orange dashed line. (a) and (b) are obtained from observed values in the validation period. (c) The modeled mean shifts are plotted against the observed mean shifts for the 12 months of the year and for the 6 dew point temperature anomalies. The modeled mean shifts were obtained using the binning model approach with normalized or anomalous data, and using all of the stations in the UCRB along with a 3-month window. One can see the observed mean shift corresponding to a $+1°C$ anomaly for the month of July (orange points and distribution from (a) and (b)) is the value on the y-axis of larger orange diamond in (c). (d) The same as (c), except using CC to predict the mean shifts.




normalized data (e.g., see Figs. 6b and 6e). In Fig. 7d, one can see all of the model predicted means, which are derived from assuming a theoretical scaling rate of 7% per °C, plotted against the observed means. Eq. 1 is used to compute the model error for the binning model using normalized data for all of the points in Fig. 7c. And similarly, one can use Eq. 2 to compute the model error when assuming a theoretical scaling rate of 7% per °C (using the points from Fig. 7d). Using Eq. 3, we can arrive

at our RMSE skill score ($\mathbf{SS}_{RMSE}$) for this case. The skill score ($\mathbf{SS}_{RMSE}$) is 0.12, with respect to the theoretical CC, for this case where the binning method is used with normalized data from all of the stations within the UCRB along with a 3-month time window. At the same time, the skill score ($\mathbf{SS}_{RMSE}$) of these model predictions with respect climatology is 0.51. Both of these skill scores are statistically significant (p<0.01). We provide a more thorough analysis of the statistical significance later in the paper.

## 4 Results

### 4.1 Performance of the Two Binning Models

Figure 8 shows model performances for the two different binning methods, with and without normalization. Figure 8a shows the RMSE skill scores of the binning model which uses the raw, or non-normalized, data. In Fig. 8a, the skill is in reference to climatology. Each individual grid cell in Fig. 8a corresponds to the skill computed where the predictions are produced

using different combinations of our spatial and temporal extents. Figure 8b shows the skill of the binning model which uses normalized data in reference to climatology. Figures 8c and 8d show the skill of the binning model when using non-normalized and normalized data, but now in reference to predictions generated using the CC scaling rate. The skill values of 0.12 and 0.51, corresponding to the example detailed in Fig. 7, can be observed in Figs. 8b and 8d at the intersection of "Basin" on the y-axis and "3-Month" on the x-axis. The skills are seen to change as a function of the spatial and temporal extents. For the

binning model which uses non-normalized data, the best model performance is achieved when data from both small spatial and temporal extents are used. This can be seen as the blue colors in the top-left corners of the Figs. 8a and 8c. In contrast, the optimal model performance of the binning model which uses the normalized data is obtained when using a temporal extent of approximatedly 3 months and a spatial extent equal to the entire UCRB. By using the non-normalized data itself, which does not correct for climatological differences in both time and space, the skill immediately begins to decrease as we use greater

spatial and temporal extents. For example, the skill in estimating the maximum daily precipitation over the period 1994-2023 at one station for the month of July will decrease, on average, when data from June or August from the same station is used and/or data from July is used from stations within a 100 km radius (see Figs. 8a and 8c, the skill decreases as one traverses to the right and/or down from the top-left cases). However, we find that this same data can be leveraged to improve the skill of the binning model when data normalization is first applied. As shown in Figure 8, the best skills for the binning model using the measured,

non-normalized, data is 0.49 and 0.08 with respect to climatology and CC, respectively. When using the normalized data with the binning model, the best model skill increases to 0.51 and 0.12, respectively. We can additionally compare these results for the same case (i.e., "Basin" and "3-Month") against the skill of the exponential model which also uses the normalized data.



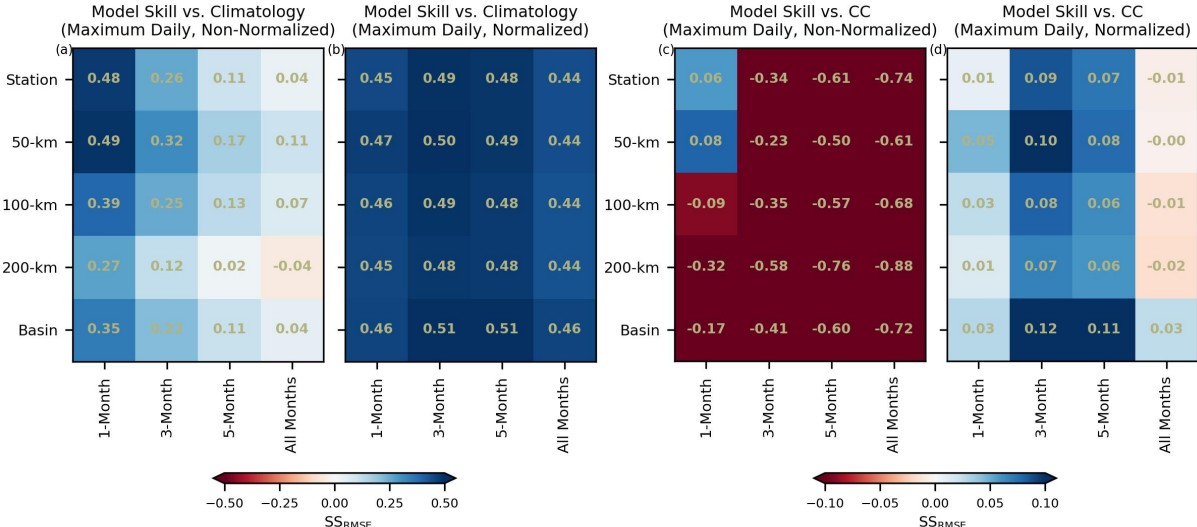

**Figure 8.** Performance of the binning model when using non-normalized versus normalized data. The performance is measured using the RMSE skill score. (a) Skill scores of the binning model, which uses non-normalized data, are shown with respect to climatology. The skill scores are reflected by the colorbar in addition to values being printed on top of the colors. Skill is seen to change as a function of making predictions using data from varying spatial and temporal extents. (b) The same as (a), but for the binning model which uses normalized data. (c) and (d) The same as (a) and (b), except that the model performance is evaluated with respect to CC.

The skill of the exponential model for that case is 0.55 and 0.18, respectively. We find that the parametric exponential model provides a statistically significant improvement (p<0.01) with respect to the non-parametric binning method.

We can further highlight why it is that the skill of the binning model, which uses the measured or non-normalized data, falls off so quickly as we further extend the temporal and spatial extents. For the UCRB, it is during the season of August-September-October that, on average, produce the most extreme precipitation events. One might decide that they want to find the maximum daily precipitation event that took place over that most extreme season (Aug-Sep-Oct) at a particular station, and they could also pair that maximum daily precipitation event with the monthly average dew point temperature for the

month of that event. However, even among months over this 3-month season, the climatological differences across time can be quite large. Consider a case where we compute anomalies of dew point temperature and precipitation using Eqs. 6 and 7, but the anomalies are computed with respect to seasonal averages. So, for each station, we take the August and October time series of dew point temperatures and maximum daily precipitation rates, and compute the anomalies with respect to a August-September-October seasonal average for that station. Then, we proceed to do the same operation for all stations. In

this way, we have removed any spatial climatological differences (because we have produced the anomalies station by station), but this approach preserves any temporal climatological differences that may exist in the data. We can plot these anomalies for the months of August and October in Figure 9a. The dew point temperature and the maximum daily precipitation anomalies





computed using seasonal averages are plotted in Fig. 9a for all of the stations and all of the years in the data set. It is clear from Fig. 9a that the dew point temperatures in the month of October are significantly cooler than those of August. They are, on average, 8.7°C cooler. While at the same time, October has extreme daily precipitation amounts that are, on average, 7% greater than those of August. If we simply take the measured data values across a season such as August-September-October and subsequently compute anomalies on that data, then the climatological differences in time will have a large impact. Due to these climatological differences between the months of August and October, for example, we would in this case underestimate the effective scaling rate. Molnar et al. (2015) and Visser et al. (2021) both showed a similar result across time at a single station. The gray dashed line in Fig. 9b, is the exponential fit, or the estimated scaling rate, of all of the scatter points in the August-September-October season where the anomalies are computed using seasonal averages (these are the data points from Fig. 9a plus the data from September). We can then compare the gray dashed line in Fig. 9b to the exponential fit of each of the individual months in this 3-month season (also shown in Fig. 9b). In Fig. 9b, we are highlighting the differences in the rate of change as a function of dew point temperature, and we therefore have all of the curves pass through 100% of normal at a 0.0°C dew point temperature anomaly. The estimated scaling rates of the three months, individually, can be seen to be very similar to one another and to what we estimate when fitting an exponential curve to all three months together using normalized data computed at the station-month level. The effective scaling rate is clearly greater than what is estimated if we had not accounted for climatological differences in time, where data values can vary substantially from month to month even at the same station. This shows how one would inaccurately estimate the scaling rate for this season simply by not accounting for the fact that October is typically cooler, but also exhibits modestly more extreme precipitation than August. Next, we can isolate the impact of climatological differences across space. To do this, we can take the measured values of the data across the stations for the single month of October. This is the same data as the blue points in Fig. 9a, though we have not yet computed any anomalies at this point. Next, we can calculate the mean over all of the stations in the UCRB and all of the years for this month of October. In Fig. 9c, we plot the scatter points for two different subregions of the UCRB, data east and west of 108° longitude, respectively. The data east of 108° longitude contains dew point temperatures 1.6°C cooler and maximum daily precipitation extremes that are 15% wetter, on average, than the data west of 108° longitude. Here, we have removed temporal climatological differences by using a single month, and we therefore isolate the climatological differences in space. The scaling rate which is fit to the scatter points of Fig. 9c is shown as the dashed gray line in Fig. 9d. Again, we observe the estimated scaling rate in gray would, in this case, underestimate the "true" scaling rate due to climatological differences in space. These examples show how important it is to remove climatological differences in both space and time. And it is because of these climatological differences, that the skill of the binning model which uses the measured data without normalization decreases so rapidly as the spatial and temporal extent increases.

## 4.2 Evaluating the Exponential Model Performance

In Figure 8, we have already shown the split-period cross-validated performance of the two different binning methods. We found that the binning method which uses normalized data produces more skillful predictions of maximum daily precipitation than the binning method which uses the measured values without normalization. Furthermore, we find that using the same



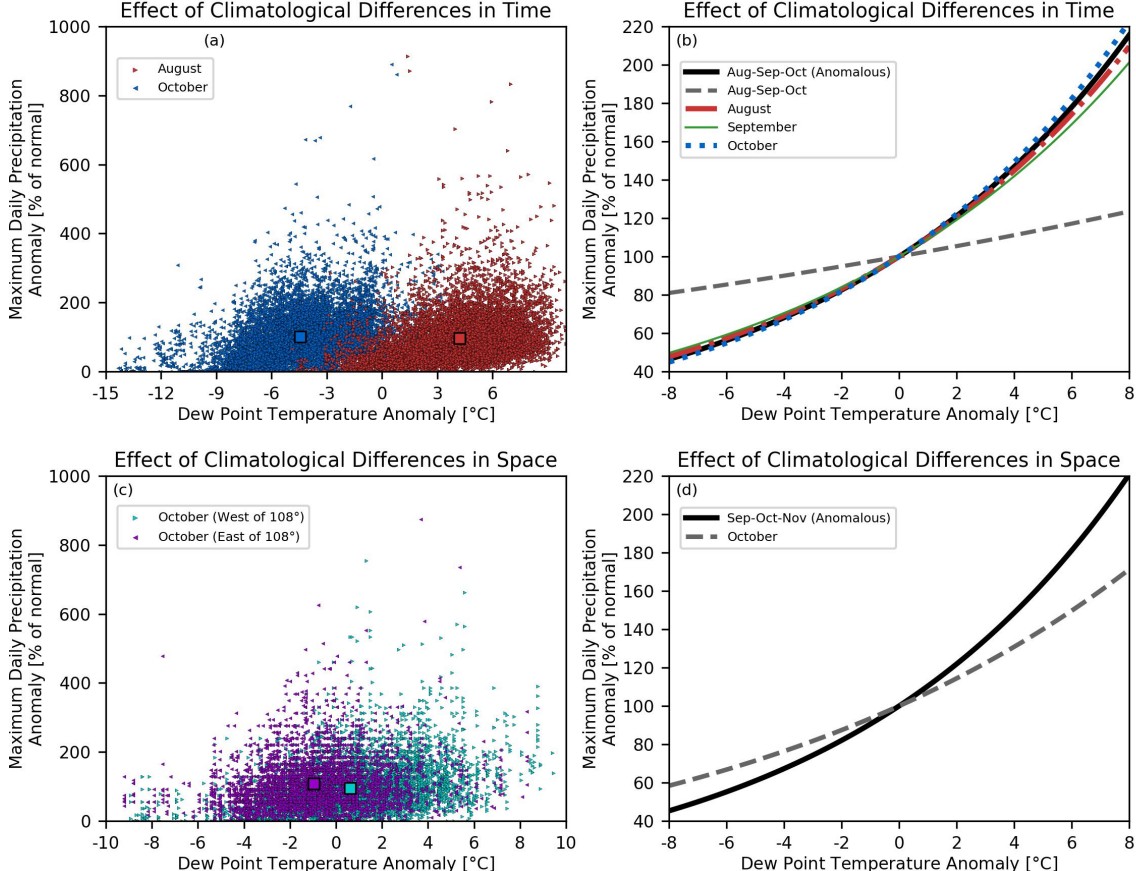

**Figure 9.** In (a), anomalies of dew point temperature are plotted against anomalies of maximum daily precipitation. For each station, anomalies are computed with respect to the August-September-October seasonal average for that station. (a) illustrates the climatological differences in time of the data. September is not shown in (a) in order to more clearly visualize the climatological differences between two differences months within a 3-month season. (b) Different scaling rates, which have been centered at $0°$C along the x-axis and 100% or normal along the y-axis, are plotted. The scaling rate obtained from the data in (a), including the data from September, is plotted as the gray dashed line. The solid black line is the scaling rate using the same data over the same season, but where anomalies are computed at the station-month level. The red, green, and blue lines are the scaling rates from each month individually. (c) is similar to (a), but now isolating the impact of spatial climatological differences. For the month of October, anomalies are computed with respect to the all of the stations across the UCRB. (c) plots the anomalies east and west of $108°$ longitude. The scaling rate fit to the values in (c) is plotted as the gray dashed line in (d), while the scaling rate for the season centered about October, using anomalies computed at the station-month level, are plotted as the solid black line.

normalized data, the parametric exponential model produces more skillful predictions than the binning method. We have shown this using a calibration period 1951-1993 and an independent validation period of 1994-2023. Here, in Figure 10, we now show the model performance for both maximum daily and maximum hourly precipitation rates. However, in Fig. 10





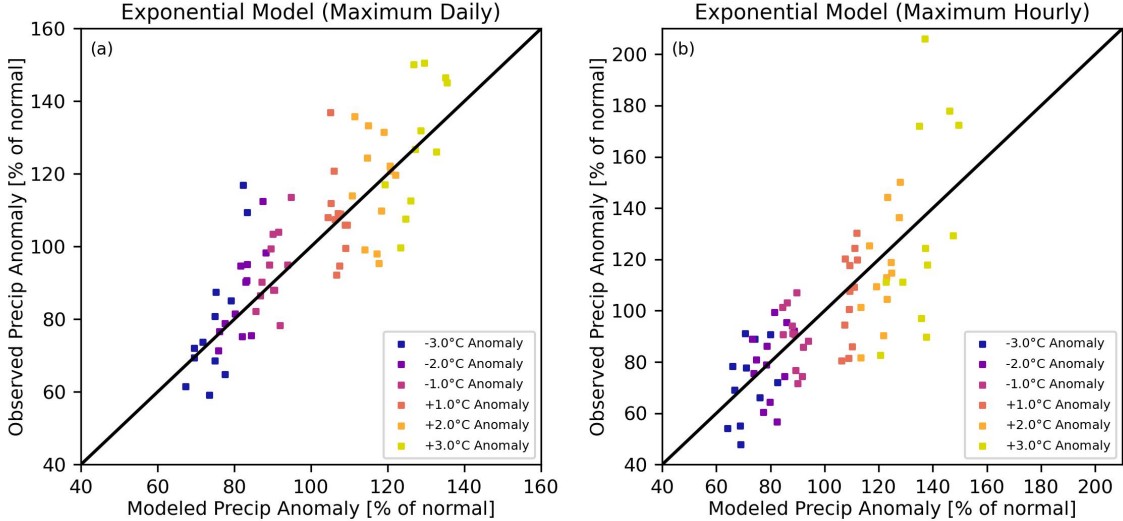

**Figure 10.** The relationship between model predicted mean shifts versus observed mean shifts for the 12 months of the year and for the 6 dew point temperature anomalies. The mean shifts are obtained from predictions of the exponential model using normalized data for all stations within the basin and a 3-month window. The modeled versus observed mean shifts are shown for the maximum daily precipitation in (a) and the maximum hourly precipitation in (b).

we use shorter validation periods. The reasons for doing this are two-fold. First, we need sufficient data in hourly dataset in order to both calibrate and validate the model. Second, we would like to illustrate the impact that different calibration and validation periods can have on model performance (the following paragraph provides further details regarding this issue). For the maximum daily precipitation, we now use a calibration period 1951-2003 and a validation period of 2004-2023. For the hourly dataset, the data can be quite sparse prior to the year 1999. Therefore, we make use of a shorter validation period to

perform an independent cross-validation for the maximum hourly precipitation. We use the period 1951-2008 to calibrate the maximum hourly precipitation exponential model, and we validate over the period 2009-2023. In both of these cases of the daily and hourly data, we have a "sufficient" amount of data to both calibrate and validate the models. Following the results concerning the best skill of the exponential model discussed above, the model is fit to the normalized data over the calibration period using all of the stations within the UCRB and using a 3-month window. Using these calibration and validation periods,

we now observe skill scores of 0.42 ($p<0.01$) and 0.35 ($p<0.01$) with respect to climatology in the model's performance of estimating the maximum daily and hourly precipitation rates, respectively. The model skill scores with respect to CC are 0.09 ($p<0.01$) for the maximum daily, and 0.12 ($p<0.01$) for the maximum hourly.

Our results show that the skill of the model predictions of maximum daily precipitation are statistically significant for both of the chosen validation cases, 1994-2023 and 2004-2023. However, the skill is seen to be quite different between these two

cases. The skill scores are 0.55 and 0.18 (versus climatology and CC, respectively) when validating over the last 30 years, and 0.42 and 0.09 when validating over the last 20 years. This leads us to another important issue that requires consideration




when assessing the skillfulness of different methods. It is worth remembering that we are dealing with extreme events that are relatively rare to begin with, and as a result, natural variability can influence our apparent level of skill in two important ways. First, the estimation of what is considered a "normal" extreme precipitation event will vary from one calibration period to another. As the length of the time series in the calibration period increases, we improve our confidence in estimating what constitutes an "average" extreme event. Consider that we want to use a station with 20 years of measurements. If we split the time period in half for calibration and validation, then we are using only 10 measurements to compute what the "average" or mean extreme precipitation is for a given station and a given month. And the second way that natural variability can influence the apparent level of skill, is that the validation must also be performed over enough cases or data points in order to obtain an accurate picture of the "true" mean shift of precipitation extremes given the different dew point temperature anomalies. To largely circumvent these two issues, we would like to predict the entire extreme precipitation array through leave-one-year-out cross-validation. Before we do that, however, we must first determine whether the mean shifts in the observed precipitation extremes, as a function of dew point temperature anomalies, are systematically changing over time. That is to say, are the observed mean shifts in extreme daily precipitation for the 12 calendar months and the 6 dew point temperature anomalies systematically moving in one direction between the calibration and validation periods? Can these mean shifts, or the scaling rates themselves, be considered stationary? Would we be making an invalid assumption that a model fit over some historic period can be applied to some "future" period?

The larger scatter points of Figures 11a-11f show the effect of the first point raised above. That point relates to the fact that the predicted mean of the extreme precipitation events can vary from one calibration period to another. The larger diamond scatter show the calculated mean shifts over the last 20 years (2004-2023) with respect to three different calibration periods: 1951-1983, 1961-1993, and 1971-2003. One can observe that the points move as the calibration period changes, which is illustrated by the fact that more than 12 diamond scatter points can be seen in each subplot. This is because what is considered "average" or "normal" changes as a function of the chosen calibration period. Next, we would like to observe if the scatter of these larger points differs systematically from randomly selected calibration and validation periods. If we observe a systematic change over time, then this would indicate that the scaling rates cannot be assumed to be stationary in time. To determine whether there is a systematic change in the observed mean shifts over time, we first generate 100 randomized 20-year calibration and 20-year validation periods chosen from only the first 53 years of data, corresponding to the years 1951-2003. For each scenario, we randomly choose two separate and independent 20-year periods, which are not necessarily sequential, from the years 1951-2003 to be used as calibration and validation. For each scenario, none of the years chosen for the calibration period are used in the validation period. We then plot the mean of the extreme precipitation anomalies for the calibration versus the validation periods for each of the 12 calendar months and the 6 dew point temperature anomalies. We repeat this procedure 100 times. We find that the cloud of these smaller scatter points generated exclusively from the data period 1951-2003, fully envelops the larger scatter points of the last 20 years. This tells us that much of the error that we observe between the scatter points and the one-to-one line in Figures 7c, 10a, and 10b can be attributed to natural variability. We do not find any discernible systematic change in the mean shifts of extreme precipitation over time. As a result, we can make a safe assumption that the P-T scaling rates are stationary with respect to time.





**Figure 11.** Influence of natural variability in evaluating performance skill. Randomly chosen 20-year calibration and 20-year validation periods are repeated 100 times. The calibration and validation periods are chosen from only the first 53 years of data, corresponding to the years 1951-2003. For each dew point temperature anomaly, the scatter points are shown between the data which could be used in calibration (x-axes) and the data which could be used in validation (y-axes). The smaller points consist of the 12 months of the year for the 100 randomized realizations. The larger diamond scatter show the calculated mean shifts over the last 20 years (2004-2023) with respect to three different calibration periods: 1951-1983, 1961-1993, and 1971-2003.



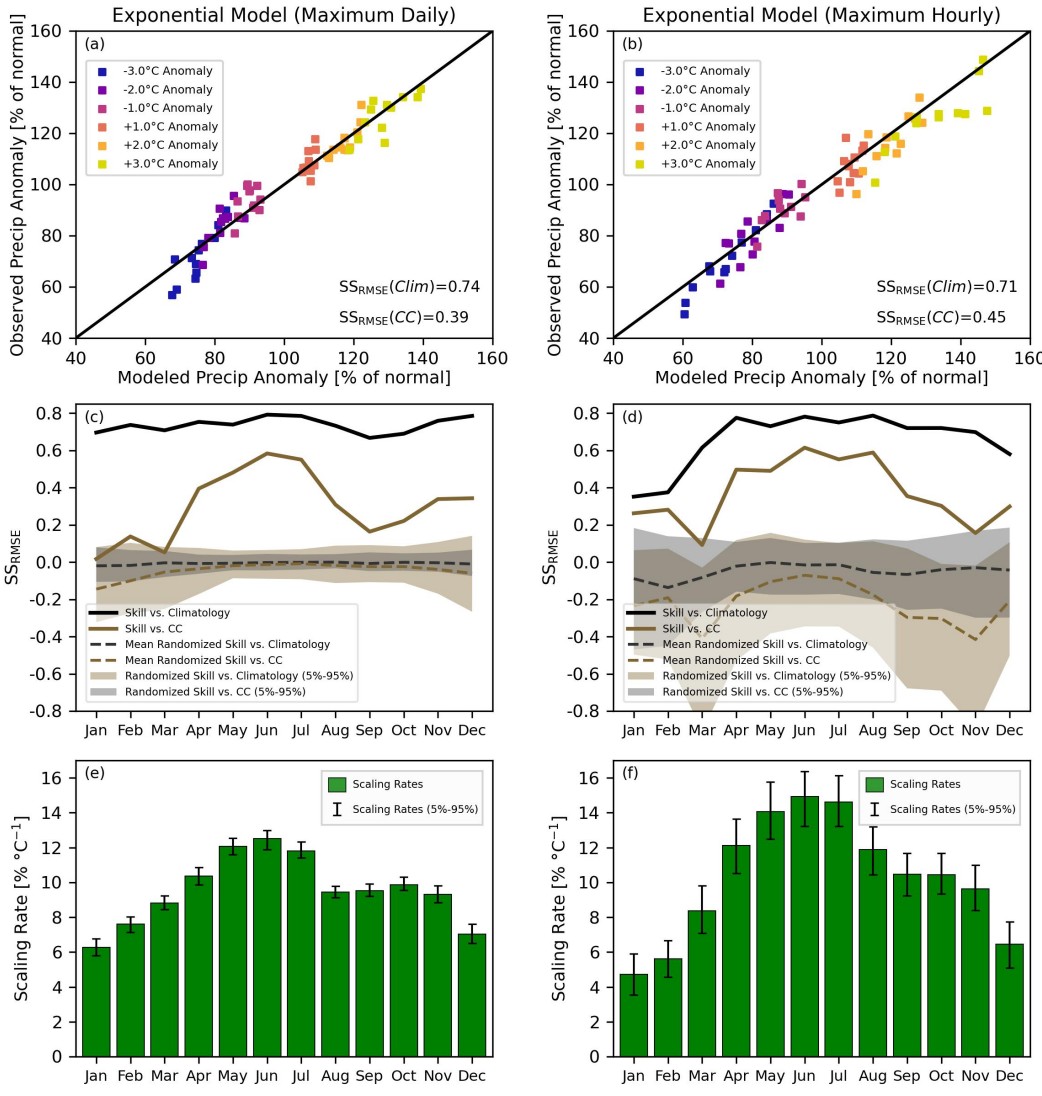

**Figure 12.** (a) Leave-one-out cross-validated modeled mean shifts in the maximum daily precipitation are plotted against the observed mean shifts for the 12 months of the year and the 6 dew point temperature anomalies. (b) The same as (a), except using maximum hourly precipitation. (c) The skill is plotted as a function of the time of year for the maximum daily precipitation predictions. The thick black and brown lines are the RMSE skill scores with respect to climatology and CC, respectively. The average skill scores of randomly generated forecasts are plotted as the thin dashed lines, with the 95% confidence interval represented by the shaded regions. (d) The same as (c), except using maximum hourly precipitation. (e) and (f) Plot the scaling rates for the different months of the year for the maximum daily and maximum hourly precipitation, respectively. The green bars are the scaling rates obtained by fitting the exponential function from Eq. 8 to the normalized data using all of the stations in the UCRB and using a 3-Month window. The 5% and 95% confidence intervals are obtained through bootstrapping.





With this assumption, we can leverage the full length of the data sets by increasing both our calibration periods and the period over which we validate. To do this, we use leave-one-year-out cross-validation to make predictions of extreme precipitation and validate over the entire period 1951-2023. For each year, the mean dew point temperatures ($\overline{\mathrm{DPT}}$ from Eq. 6) and the

mean extreme precipitation ($\overline{\mathrm{P}}$ from Eq. 7) are computed using only the calibration years which exclude the year for which we are going to make model predictions of the anomalous extreme precipitation. Likewise, the exponential model is fit to the data from the calibration years, and model predictions are produced for the given validation year. For each validation year in turn, we also retain the observed precipitation anomalies computed using mean precipitation calculated from the current set of calibration years. Then, we step to the next year until we have covered the entire time period, 1951-2023. In Figures 12a and

12b, we plot the modeled versus the observed mean shifts in the maximum daily and hourly precipitation rates as a function of the 12 calendar months and the 6 dew point temperature anomalies. These mean distributional shifts are the average of the data anomalies which have been computed through the leave-one-year-out cross-validation. We observe that the skill of the predictions increases with respect to our split sample validation. This is because we use more data to calculate our means in Eqs. 6 and 7, and more data points on which to validate. Now, our skill scores with respect to climatology are 0.74 (p<0.01)

and 0.71 (p<0.01) for predicting maximum daily and maximum hourly precipitation, respectively. Similarly, our skill scores are 0.39 (p<0.01) and 0.45 (p<0.01) with respect to CC (see Figs. 12a and 12b). We can further disaggregate the skill, and plot it as a function of calendar month. In each month, for example, we use the predicted versus observed 6 dew point temperature anomalies to compute the corresponding skill scores. We use the same modeling framework to generate 100 randomized simulations to test the statistical significance of the skill as a function calendar month. For each of the simulations, randomly

selected precipitation anomalies are chosen from the calibration period and used as the model predictions. Each set of the 100 randomized simulations of predicted precipitation anomalies are evaluated against climatology in order to establish statistical significance. The randomized predictions are also superimposed on top of the theoretical CC relationship, at the given dew point temperature anomalies, in order to evaluate the statistical significance of the skill scores with respect to the theoretical CC scaling rate. We show in Figs. 12c and 12d that the skills of our proposed methodological approach are statistically

significantly (p<0.05) better than climatology for all calendar months for both the maximum daily and maximum hourly precipitation anomalies. At the same time, our proposed methodology performs statistically significantly better (p<0.05) than the theoretical CC scaling rate for nearly all months at both the daily and hourly resolutions. It is only for the maximum daily data, and for the months of January and March, that the exponential model fit does not statistically significantly outperform the theoretical CC scaling rate. What this tells us, is that the scaling rates at this time is near CC, and that during instances such

as these, we cannot necessarily provide model predictions that are an improvement over assuming the theoretical CC scaling rate. The scaling rates for maximum daily and maximum hourly precipitation are plotted in Figs. 12e and 12f as a function of calendar month. The scaling rates for both the maximum daily and maximum hourly precipitation are observed to be higher during the summer months than they are for the winter months. We find that the spread between the summer and winter scaling rates is greater for maximum hourly precipitation than it is for maximum daily precipitation.





## 5 Conclusions


Using data from the Upper Colorado River Basin, this study begins by illustrating some of the most common challenges in estimating P-T scaling rates. Our aim, herein, has not been to provide a comprehensive overview of every existing methodology related to P-T scaling rates, but rather to focus on some of the prevailing challenges confronting scaling rate estimation along with some proposed solutions. We find three primary challenges that require careful consideration prior to implementing any

estimation of scaling rates. These are: 1) using data across multiple stations and/or months, without normalization, can lead to inaccurately estimating the scaling rate due to climatological differences that exist in both space (from station to station) and time (from month to month), 2) differences in sample sizes, and 3) the temporal independence of the data. Applying a binning method with non-normalized data which is pooled from multiple stations and/or months fails to address all three of these problems. The methodology proposed by Zhang et al. (2017) resolves many of these problems by using the normalized

values of maximum precipitation accumulations and dew point temperatures which are found over a seasonal window. Their approach effectively deals with the second and the third problems, while only partly addresses the first problem. In this paper, we have shown that even climatological differences across a season can lead to an inaccurate estimation of the effective scaling rates. In particular, we showed in the case of the UCRB that the average dew point temperatures in August are more than 8°C warmer than October, while at the same time the average extreme precipitation in August is slightly less than that of October.

Without first normalizing the data at the station-month level (where anomalies are computed station by station and month by month), these climatological differences across time leads to an underestimation of the scaling rate in the UCRB for the August-September-October season.

Building on the work of Zhang et al. (2017), we advocate for estimating scaling rates using normalized data, or data anomalies, at the station-month level. Using normalized data allows us to more effectively leverage data from multiple stations and

across multiple months. By using the normalized data, we circumvent all three of the problems we raise in the first part of our paper. Then, we use different methods to estimate P-T scaling rates over different calibration periods, with and without the normalized data. Next, we go a step further and use these estimates to make cross-validated predictions of extreme precipitation, at the station-month level, as a function of dew point temperature. The performance of the predicted mean shift in the distributions versus the observed shift is subsequently evaluated at different dew point temperature anomalies between

-3°C and +3°C. Our results clearly show the value of normalizing the data prior to estimating any scaling rates and producing predictions of extreme precipitation. The performance of a binning model which uses raw, or non-normalized, data is seen to quickly degrade as stations are pooled from further than 50 km away and/or when adjacent months are used (e.g., using data in June and August to predict July). In contrast, the skill improves when using a binning model with normalized data which has been pooled across a basin/region and across a 3-month window. We find that the predictive skill is further enhanced when

using a parametric exponential model fit instead of a binned model. The predictions from the exponential model are found to be statistically significantly more skillful than either climatology or assuming a theoretical CC scaling rate.

Lastly, we show the estimated scaling rates in the UCRB across the twelve calendar months for both the maximum hourly and maximum daily precipitation extremes. The scaling rates in this region are observed to exhibit seasonality, with the scaling



rates in the winter months being below the theoretical CC rate of 7% per °C and the scaling rates in the summer months being more than double CC. Additionally, we find the variability of the scaling rates between the summer and winter months to be greater for maximum hourly precipitation than it is for the maximum daily precipitation. In this study, we have achieved skillful cross-validated predictions of extreme precipitation in the UCRB by using normalized data in a parametric exponential model. With this knowledge in hand, we can continue to improve our understanding of how P-T scaling rates vary across different months in other regions. Furthermore, one can evaluate how skillfully those rates can be applied in a changing climate.

*Data availability.* Supporting data can be found at https://doi.org/10.6084/m9.figshare.29858954.v1 (Switanek et al., 2025).

*Author contributions.* The study was conceived by Matthew Switanek. All code, analysis, and figures were produced by Matthew Switanek with input from all coauthors. The original draft was written by Matthew Switanek with assistance from all of the other coauthors.

*Competing interests.* The authors do not have any competing interests.

*Acknowledgements.* The authors want to thank the California Department of Water Resources (contract number: 4600015149) and the University of Graz for funding this research. The lead author would additionally like to thank Andreas Prein for some early discussions pertaining to this research.



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
