# Peer review of "Precipitation-temperature scaling: current challenges and proposed methodological strategies"

_EGUsphere, 2025_

## Author Comment (AC1)

**Reviewer 1:**

Let me start by stating the manuscript is well written and I believe worthy of publication. However, I am concerned that the novelty has been misrepresented. The main issue here is the authors provide 3 shortcomings they have overcome building on Zhang et al (2017). I do not agree with this statement as I believe each individual shortcoming has been addressed in previous literature. Here the novelty lies in combining the approaches from three different manuscripts. I believe this combining of methods is a worthy contribution and an important one.

The authors would like to thank the reviewer for their time and effort in providing useful feedback concerning our paper. In line with the reviewer's point, we will endeavor to make changes to our paper to better position our work with respect to prior research/literature. Also, we will reframe the novelty of tackling each individual shortcoming but rather clarify that it is the combination of the three shortcomings that we attempt to respond to in a pragmatic and systematic manner. We appreciate that the reviewer acknowledges this general aim as important and relevant.

I have two major comments which I hope the authors can address.

Major comment 1.

I strongly believe that data for P-T scaling should not be pooled without standardising – and agree with the authors, but this was demonstrated in Visser et al (2022) and Molnar et al (2015). The authors have cited these papers in their manuscript and to their own admission at line 379 they state their work bears a strong resemblance to Visser et al (2022) and Molnar et al (2015) so why not make this point in the introduction that they build on these authors work?

We will aim to provide in our revised paper a better review of where our study fits in with what has already been done. We will add references to these studies, which we build upon, earlier in the manuscript.

Further, Figure 2: A standardisation was already proposed in Visser et al (2022) and I quote in their introduction "We introduce standardized pooling...". This should be acknowledged here.

Thank you for this point. We will better acknowledge the standardization that was performed in Visser et al. (2021). In their paper, they do standardize by subtracting the mean and dividing by the standard deviation. They do this for each station. However, they do not apply the method month by month. As a result, they control for some climatological differences across space, but not in time. Additionally, we apply a different approach to the normalization, whereby we produce percent of normal (or average) precipitation values and degrees from normal (or average) dew point temperature. However, we agree that we

need to better acknowledge, for example, the work done by Visser et al. (2022) and highlight our adaptations to their approach in a clearer manner.

Figure 3 and 4: The issue that bins at the extremes have less events, and some binning techniques don't consider independence were both points made in Wasko and Sharma (2014) and hence quantile regression using independent events was proposed. This point also relates to Line 406. This should be acknowledged here.

Thank you for pointing out this issue. We had presented and used the binning method as many previous and current studies continue to use it. In our revised paper, we will shift to using quantile regression instead of the binning method to illustrate some challenges a researcher faces when estimating P-T scaling rates. Additionally, quantile regression will be used to provide an additional reference or benchmark scaling rate estimate in the Upper Colorado River Basin.

Figure 5: The use of a monthly (or seasonal) temperature was proposed by Zhang et al (2017). This should be acknowledged here.

In our paper, we had already acknowledged that Zhang et al. (2017) uses seasonal data. However, we will add in an additional credit to their work in the revised paper.

In sum, while the justification of the proposed methodology presented in this manuscript is much more elaborate than previous manuscripts (and hence I am a proponent of it being published) the framing needs to change I believe to duly pay respect to the previous research. The method proposed here is more a combination of methods proposed by Visser et al (2022), Wasko et al (2014), and Zhang et al (2017) and the introduction and conclusion should be restructured accordingly.

Thank you for this assessment. We will work to better acknowledge the great work that has already been done by these authors and highlight our own addition to it by combining and expanding it.

**Major comment 2.**

It is odd that the authors choose 7% per degree as their truth when calculating the skill, when by their own admission in Figure 12 the scaling is not aligned with CC? In some way this should addressed, with at least more focus on the actual scaling rates. The reason is – these are empirical relationships, without a "truth".

In this paper, we are building to the result that scaling rates need to be considered different between regions and across different seasons. It is precisely the methodology that we propose in this paper, where we suggest estimating different scaling rates for each month of the year across a region. Our method is

the act of computing scaling rates which rely on temporally independent data which have been normalized station-by-station and month-by-month. Furthermore, we propose only using data within a certain seasonal window, or up to a certain spatial distance, to fit the model. Doing this gives different results, with better model performance, than fitting the model to all of the data throughout the year. This approach of normalizing the data and only using data within a certain spatial and temporal window is central to our proposed methodology. The scaling rates in Figure 12 are the result of our proposed methodology, and if we were to compare model performance against the scaling rates from Figure 12, then we end up evaluating model performance against itself. That said, we will also include in our revised paper a benchmark model that relies on a scaling rate which is fit to all of the non-normalized data from the Upper Colorado River Basin (UCRB). We will, therefore, compare to 1) climatology, 2) a 7%/deg (theoretical benchmark), and 3) a benchmark scaling rate specific to the UCRB. That way, we can compare to an additional benchmark reference model which is specific to the UCRB, but it is not implementing the very approach which is central to our paper.

**Minor comments:**

Title: The title suggests a review and noting that some of the current challenges have been resolved the title could be amended.

We thank the reviewer for this comment. Our revised paper will include a new title that more accurately reflects our work.

Line 1-2: Does sub-daily rainfall scale at 7%? There is now much review/meta-analysis work showing it is likely higher? e.g. Fowler et al (2021); Wasko et al (2024). The IPCC reports also point to higher than 7% scaling for sub-daily rainfall.

We are comparing against the theoretical Clausius-Clapeyron, which to our understanding does not change as a function of temporal scale (e.g., sub-daily to daily). However, as we stated above, we are going to provide another benchmark scaling rate for the UCRB using both hourly and daily data which has not been normalized.

Line 60 onwards: The point of pooling resulting in "incorrect" scaling was well made in Molnar et al (2015) and has been made in papers by Berg and Haerter – making the point that a lot of this has to do with different storm types, but this was never mentioned here?

One can view the use of normalization, station-by-station and month-by-month, as a proxy for different storm types. Molnar et al. (2015) stated, "the scaling rates for all events are systematically higher than those of the individual lightning and no-lightning subsets because of the mixing of stratiform events at

low temperatures and convective events at high temperatures." They are pointing to the fact that different seasons experience different types of storms. We agree that this is a reason why the raw, non-normalized data should not be mixed. We will add that the normalization also helps us address different underlying storm types that are predominant in different times of the year.

Line 113: "incorrect" is a strong word when we don't know the truth, scaling's are correlations and they're all true in some way regardless of the method.

Indeed, we agree that the word "incorrect" might be too strong for what we try to address. We will adopt the wording to both make clear that predictability depends partly on the method used without attempting to judge on absolute correctness of one method versus another.

Figure 8 nicely presents that the pooling of standardized data works, but Figure 8d also shows that monthly data can be safely pooled after standardization and the performance is similar (Column 1 vs Column 4) – could this point me made in the text?

Thank you for this point. We can improve our discussion concerning this point in the text.

**References:**

Berg, P., Haerter, J.O., 2013. Unexpected increase in precipitation intensity with temperature — A result of mixing of precipitation types? Atmospheric Research 119, 56–61. https://doi.org/10.1016/j.atmosres.2011.05.012

Fowler, H.J., Lenderink, G., Prein, A.F., Westra, S., Allan, R.P., Ban, N., Barbero, R., Berg, P., Blenkinsop, S., Do, H.X., Guerreiro, S., Haerter, J.O., Kendon, E.J., Lewis, E., Schaer, C., Sharma, A., Villarini, G., Wasko, C., Zhang, X., 2021. Anthropogenic intensification of short-duration rainfall extremes. Nature Reviews Earth & Environment 2, 107–122. https://doi.org/10.1038/s43017-020-00128-6

Molnar, P., Fatichi, S., Gaál, L., Szolgay, J., Burlando, P., 2015. Storm type effects on super Clausius–Clapeyron scaling of intense rainstorm properties with air temperature. Hydrology and Earth System Sciences 19, 1753–1766. https://doi.org/10.5194/hess-19-1753-2015

Visser, J.B., Wasko, C., Sharma, A., Nathan, R., 2021. Eliminating the "hook" in Precipitation-Temperature Scaling. Journal of Climate 34, 9535–9549. https://doi.org/10.1175/JCLI-D-21-0292.1

Wasko, C., Sharma, A., 2014. Quantile regression for investigating scaling of extreme precipitation with temperature. Water Resources Research 50, 3608–3614. https://doi.org/10.1002/2013WR015194

Wasko, C., Westra, S., Nathan, R., Pepler, A., Raupach, T.H., Dowdy, A., Johnson, F., Ho, M., McInnes, K.L., Jakob, D., Evans, J., Villarini, G., Fowler, H.J., 2024. A systematic review of climate change science relevant to Australian design flood estimation. Hydrology and Earth System Sciences 28, 1251–1285. https://doi.org/10.5194/hess-28-1251-2024

Zhang, X., Zwiers, F.W., Li, G., Wan, H., Cannon, A.J., 2017. Complexity in estimating past and future extreme short-duration rainfall. Nature Geoscience 10, 255–259. https://doi.org/10.1038/ngeo2911

---

## Author Comment (AC2)

**Reviewer 2:**

This paper proposes an integrated approach to improve our estimates of precipitation-dew point scaling rates. The idea relies on the seasonal normalisation of dew point (additive normalisation) and monthly maxima of precipitation at daily/hourly scales (multiplicative normalisation).

Overall, the manuscript collects a set of ideas from literature (although sometimes with incomplete referencing), and proposes an approach to integrate those ideas. For this reason, I think the title is slightly misleading, as it tends to overgeneralise the breadth of the contribution and oversell the novelty.

The authors would like to thank the reviewer for their time and effort in providing useful feedback concerning our paper. We will try to improve our referencing and clarify our contribution. Our revised paper will include a new title that more accurately reflects our work.

Before commenting, I'd like to say that I found the manuscript difficult to follow, so some of my comments may be associated with misunderstandings. I'll be happy to discuss them further.

I think the methodological idea has potential, but the implementation is rather convoluted and it hinges on some subjective choices not fully motivated in the text. Also, I am not convinced by the validation that, to my understanding, is based on mean deviations - while in precipitation-temperature scaling we are typically interested in extremes.

Thank you for the comment. We will work to improve the readability and the clarity of the revised paper. To be clear, we are using extremes throughout the paper. Please refer to our response, found below in this document, corresponding to your point at Line 308 in the original paper version.

Overall, the study has potential, but I think major work is required before it can be reconsidered for publication.

I provide below here my specific comments, in the order they appear in the manuscript.

☐ Line 7-8: here it is not yet clear what 'normalised' means. I suggest to briefly explain it

We can add a brief description earlier in the paper. At this point in the abstract, we can rewrite the text starting at line 7 in the paper to read:

"Specifically, we compare multiple approaches, including those using raw (non-normalized) and normalized data (using % of normal for precipitation and °C from normal for temperature or dew point temperature), to estimate P-T scaling for hourly and daily extreme precipitation."

| Line 20: is looks odd to have two significant digits for the 1 degree and not for the 7% (which is approximated)  Thank you for this suggestion. We can adjust the number of significant digits to be consistent throughout the manuscript.                                                                                                                                                                                                                                                                                                                                                                                                                                                                                                                                                                                                                                                                                                  |
|----------------------------------------------------------------------------------------------------------------------------------------------------------------------------------------------------------------------------------------------------------------------------------------------------------------------------------------------------------------------------------------------------------------------------------------------------------------------------------------------------------------------------------------------------------------------------------------------------------------------------------------------------------------------------------------------------------------------------------------------------------------------------------------------------------------------------------------------------------------------------------------------------------------------------------------------|
| Line 20: Extreme precipitation events depend on other variables, not only moisture in the column. I suggest to slightly rephrase.  This is a valid point, and we will rephrase this sentence to include other variables which also influence extreme precipitation.                                                                                                                                                                                                                                                                                                                                                                                                                                                                                                                                                                                                                                                                          |
| Line 42: Technically, dew point is defined as "the temperature at which air saturates when cooled at constant pressure" (e.g., see wikipedia or any atmospheric sciences book). It follows that dew point over a given location contains information only on the available moisture, and not on temperature. In fact, knowing the dew point and the pressure, it is not possible to calculate the temperature. For this reason, also the sentence in lines 49-50 needs to be updated ("the chosen temperature variable" should be changed to something like "the chosen variable").  Thank you for this suggestion. We will update our definition of dew point temperature. Many prior studies which have focused on scaling rates have compared air temperatures and dew point temperatures. It is common practice to refer to dew point temperature as a temperature variable. However, we can more simply refer to the "chosen variable". |
| Line 45: Marra & al 2024 do not use the binning method, please remove the reference. The reference instead can be relevant to the sentence at lines 65-67 and 67-68.  Thank you for pointing this out to us. We will change the references accordingly.                                                                                                                                                                                                                                                                                                                                                                                                                                                                                                                                                                                                                                                                                      |
| Line 54: data normalisation has not been defined. What do you mean by that? I expect to learn it later, but it should be explained earlier. Good point, we will provide a description in the text, that we then later define mathematically.                                                                                                                                                                                                                                                                                                                                                                                                                                                                                                                                                                                                                                                                                                 |
| Line 69: you seem to use the binning method. Indeed the introduction mentions this method, but does not mention there are some alternatives, namely the quantile regression, which is known to provide more robust estimates (if used properly) and to require much less subjective choices. This becomes more relevant given the fact that later you use an exponential model for the means and that the evaluation is done on mean                                                                                                                                                                                                                                                                                                                                                                                                                                                                                                         |

values. I believe quantile regressions could help you solve both these issues.

Thank you for this comment. In our revised paper, we will shift to using quantile regression instead of the binning method to illustrate some challenges a researcher faces when estimating P-T scaling rates. Additionally, quantile regression will be used to provide an additional reference or benchmark scaling rate estimate in the Upper Colorado River Basin.

☐ Line 75: the introduction fails to mention the literature that investigated the impact of process heterogeneity on the emerging scaling rates (e.g., Molnar et al 2015, which is cited but for other reasons, or De Silva & al 2025 Nat Geo). Given the focus on seasonality, this is a critical aspect that needs to be addressed. For example, is the normalisation handling the same problems? Is it only an approximation of what a classification would do in a more proper manner?

We aim to provide one way to improve the accuracy of scaling rate estimates, and how they can skillfully provide predictions of extreme precipitation. We have used normalization to remove climatological differences found in the data. Classification will group data that are statistically similar, while we have transformed all of our data so that all of the data exhibit similar statistics by having a common frame of reference. We have already shown, and will aim to make more clear in our revised paper, that our approach is statistically significantly skillful. We will additionally make sure to add in these points and references in the Introduction. Perhaps in some future study, others can apply an adjusted version of our proposed methodology. However, that is beyond the scope of our work here.

☐ Line 108: "make skillful predictions of extreme precipitation". It is not yet clear what you mean by "predictions". What are you trying to predict? is it the scaling rate for a given place and month? Is it the precipitation magnitude associated with a given probability (percentile) at a given temperature? Needs to be clear right from here, otherwise it is difficult to follow.

In our revised paper, we will try to be more clear about what exactly we are predicting. We will also change some of the terminology in our revised paper. Notably, we will use commonly defined index names to refer to our maximum hourly and daily precipitation values, which are obtained at each station and each month. These will be referred to as Rx1h and Rx1day. Consider that we are trying to predict daily maximum precipitation (Rx1day) given changes in monthly dew point temperature. Using crossvalidation, we predict, for each station and for each month, how much above or below normal the Rx1day would be given the dew point temperature anomaly at that station and in that month. For example, let's say that some station in July has a mean Rx1day across the calibration years of 15 mm, and the mean dew point temperature is 10°C. Through our model fit in the calibration period, let's say that Rx1day scales by 12%/deg. Now, if that station had a dew point temperature value of 11°C (i.e., 1°C above normal) in one year in the validation period, we would predict that the precipitation would be 12% greater than average (i.e., 112% of normal), or 16.8 mm. We perform the same procedure for all stations and months in the validation period. So, we are predicting how much the intensities of Rx1h and Rx1day are changing as a function of dew point temperature anomalies. We then evaluate these predictions versus observations, in the validation period, across different bins of dew point temperature anomalies. We evaluate model performance using deterministic values, corresponding to a linear exponential model fit. In our revised paper, we will also add some discussion about the utility and interpretation of deterministic versus probabilistic predictions.

☐ Fig. 2 and the related analyses. Technically, the hook structure could be created by lack of sufficient observations to properly estimate rare percentiles (Marra & al 2024). What the normalisation does is to remove the heterogeneity. The result is that the sample is homogeneous and its portion at high temperatures becomes more populated, allowing for a better estimate of the large percentiles. Therefore, the normalisation alone is not the "cure" to the hook structure, it also needs sufficient data sample. I suggest to better state this.

This is an interesting comment. Indeed, increasing the sample size will have an impact. However, in the case of our data in the Upper Colorado River Basin, we did not change the data sample. In each case, using non-normalized versus normalized data, we took the average of the data within a bin only when at least 10 points fell in the top 0.1% or 1%. So, in the case of the UCRB, it is not the difference in sample sizes that explains the hook, but rather climatological differences in space and time. After normalizing the data, that alone was enough to remove the hook structure. We will restructure and restate this point in the revised version of our paper.

| Lines 198-205: this is a lot of text to say "precipitation is stochastic" |
|---------------------------------------------------------------------------|
| We will attempt to simplify the text here.                                |

☐ Lines 209-210: this resembles some of the ideas behind Marra et al. 2024

Thank you for this suggestion. Instead of collocated temperature values, they use daily-averaged values of temperature. We aggregate the dew point temperature over a longer period of time (i.e., the month). Indeed, we agree that it is good advice to relate our results to their paper here.

☐ Line 236: why only the mean is included in the normalisation? Doesn't the variance also count? It should be stated something about the assumption behind this normalisation.

There is not a precise, fixed definition of the terms normalization and standardization. The authors view standardization as the process of computing z-scores (because the data is being standardized by subtracting the mean and dividing by the standard deviation). We have interpreted and presented normalization as the process of adjusting values to a common scale, where the data can then be more easily compared and contrasted with a common frame of reference. Ultimately, we want to provide scaling rates as %/°C. Therefore, this informed us as to how we wanted to scale the data. We wanted the precipitation data to be "% of Normal" and dew point temperature data to be "°C from Normal". Then, after applying those normalizations, we can easily fit a model which provides us with a scaling rate as %/°C. This is not the case if we were dealing with z-scores, as those values are unitless, and would thus not provide easily interpretable scaling rate values such as % change of Rx1day precipitation per degree change in dew point temperature.

☐ Line 246-246: you repeatedly claim that the normalisation above "effectively timely removed the three common challenges". This needs to be shown, for example you can compare the distributions before and after for some example cases.

We show this to be the case for the temporal independence of the data. The normalization of precipitation and temperature does put the data on similar scales for each respective variable. As stated above, we want to fit and estimate scaling rates as %/°C. To do this, our normalization procedure does not perfectly put all of the data to the same distribution. Some station could have greater skewness in its summer precipitation than its winter precipitation. Similarly, the variance of dew point temperatures at some station could be greater in the winter than for the summer. We don't perfectly collapse all of the data to one common distribution, but we have a common frame of reference that is then useful in providing scaling rates as %/°C. We can more carefully state that while we do not entirely remove these challenges, we do show how useful the normalized data is in improving our predictions of extreme events.

☐ Line 251-252: "any reference..." I suggest to include this part, in some way (perhaps referring to section 3.2), much earlier in the text. Good point, we will bring up earlier in the text what we mean by normalized data. ☐ Line 294: I don't understand how this is possible. Perhaps the way daily and hourly maxima are defined is not sufficiently clear? As we state in the Data section 2, the hourly and daily data are different data sets. For the hourly dataset, there are not as many stations and the average period of record is shorter than with the daily dataset. ☐ Line 308 and Figures 6,7,8,11,12: if I understood correctly, the evaluation is done on the mean values of the bins, and not on the extremes. Is this correct? I don't understand how is this useful for extremes, which are the target of P-T scaling applications. I think more reasoning should be provided here. All of our precipitation data used for all of those figures are extreme precipitation data. To clarify, we can provide a comparative example. Let's take the traditional binning method with data pooled from all months and stations within a region such as the Upper Colorado River Basin. Next, we could take bins of dew point temperature and find, for example, the hourly precipitation amounts in the top 0.1% in each bin. If one were then to take the average of these points that fall within the top 0.1%, we are still dealing with extreme events but we are finding the average extreme precipitation amount across each bin. In any distribution, if you were to take the top 1% or 0.1% of the values in that distribution, and then average those values, you are providing information about the extremes or the tail end of the distribution and not the mean of the entire distribution. Now, instead of using the binning method, we take the hourly maximum precipitation for each month and each station. We will try to improve our content in the paper revision as to why this is a good idea. Then, we perform our modeling and evaluation over these extreme precipitation values that fall in the top  $\sim 0.1\%$  (for hourly data) and  $\sim 3.3\%$  (for daily data). ☐ Fig 7 and several other results/validation: why are 2 degree C bins used? What is the sensitivity of the outcomes to this choice? Thank you for this comment. We plan to include a more thorough analysis of the statistical significance of our model and its sensitivity to evaluating

the model performance when using different bin sizes.

- □ Fig 7 and 10: usually the reference value (observed in our case) is plotted in the x-axis to facilitate interpretation (model overestimation is above the 1:1 line, and underestimation below)
  While we appreciate the reviewers observation here, it is our preference to plot the results the way that they have been presented. We have clearly labeled the x- and y-axes, to help facilitate the reader making sense of the results.
- □ Line 460-461: I agree on this consideration, but I wonder how much statistically robust it can be considered. For example, the plots for -3 and +3 (Fig 11a,f) show quite many large dots at the boundaries of the distribution. Perhaps a statistical test can help on this. Perhaps the same Montecarlo used here can be used, if many more samples are generated? We respectfully disagree with the reviewers interpretation of this figure. The vast majority of the larger points fall well within the range of the smaller points. While a few of the larger points do fall near the edges of the cloud of smaller points, this is also to be expected with randomly generated data. We do not find the existence of more larger points that are near the boundaries than one would find with randomly generated data.
- ☐ Lines 492-494: I don't understand how this finding relates with the finding that normalised values allowed for putting different months together (e.g., fig 9). Isn't this result suggesting that we should not mix months even with normalisation? Please explain.

Thanks for this comment, and we acknowledge that this point requires additional explanation. Our results from Figure 8 show that normalizing allows us to leverage data across space and time in a way that nonnormalized data does not allow. What Figure 9 also shows is that the way we normalize matters. Even data across the same stations in the same season, for example, can be influenced enough by climatological differences to where we inaccurately estimate the underlying scaling rate (Figs 9a,9b). These results tell us that if we obtain maximum precipitation amounts at each station across a 3-month season, and then normalize those values, we encounter a problem with climatological differences across time. We showed how different the climatologies are between the months of August and October. Taking an extreme value across the August-October season will end up mixing statistically significantly different climatologies and will adversely affect the estimation of the scaling rate. However, if we normalize the values for each station and each month first (across all Augusts for a single station, for example), then this effectively allows us to

better leverage the data across months in a season, as depicted in Figure  $8. \,$

---

## Author Comment (AC3)

**Reviewer 3:**

The paper presents a relevant and interesting idea. Normalizing station-month data to reduce artefacts in precipitation-temperature scaling is sensible, and the exponential model fitted to normalized anomalies improves predictive skill in the Upper Colorado River Basin. However, several aspects of the analysis and interpretation need tightening before the conclusions can be considered robust.

The authors would like to thank the reviewer for their time and effort in providing useful feedback concerning our paper. We will work to improve our analysis and interpretation in our revised paper.

The main issue is the lack of discussion on data quality control. The paper doesn't explain how precipitation observations were checked or filtered. Since Ali et al. (2022) highlights errors from coarse measurement precision and faulty readings, this needs to be addressed directly in the data section, with a brief note on the checks used or potential uncertainties.

The hourly data was checked and quality controlled against the daily data set, where both have been aggregated to a monthly resolution. We will include a few additional lines outlining this process. We can also include an additional figure in an appendix that shows the how the scaling rates differ when using a precipitation resolution of 0.1mm versus 1.0mm.

The temperature binning method could also be improved. Fixed temperature intervals cause uneven sampling-cooler bins dominate while warm bins remain sparse. Using bins with roughly equal numbers of data pairs would produce more balanced estimates.

We agree completely that the mentioned points are common problems with the binning method. In our revised paper, we will shift to using quantile regression instead of the binning method to illustrate some challenges a researcher faces when estimating P-T scaling rates. Additionally, quantile regression will be used to provide a reference or benchmark scaling rate estimate in the Upper Colorado River Basin.

Normalization is useful for removing spatial and seasonal effects, but it can hide genuine long-term trends. Subtracting historical station-month means risks erasing real climate signals in dew point or rainfall. The assumption of stationarity and the leave-one-year-out validation don't fully test for this. If the data are non-stationary, the resulting scaling estimates may be biased.

Thank you for this point. We use Figure 11 to show that we find no evidence of non-stationarity or systematic changes in scaling rates over time. However, we can add a note of caution about interpreting the results, given that the scaling rates may indeed be non-stationary.

From a statistical standpoint, a hierarchical model would be more robust than treating all stations equally or independently. It would allow shared information across stations while preserving local variations. Alternatively, quantile regression or a generalized additive model could capture nonlinear relationships without relying on arbitrary bins, and would better describe high-end percentiles.

Thank you for this comment. To reiterate our point from above, we will shift our content in the revised paper to implement quantile regression instead of the binning method.

The fitted exponential coefficient also needs clearer interpretation. The slope parameter b is treated as "% per °C," but the correct expression is (exp b - 1). Using b directly can slightly misstate the scaling rate.

This is a good point. We will change the exponential function to more explicitly have it to be interpreted as "% per °C". This can be done with a model taking the form  $y = ax^b$ , where b are the monthly dew point temperature values, a is a multiplicative offset, and x provides information concerning the scaling rate. For example, assume a=1, b=2 and x=1.08, then  $y=1*(1.08)^2=1.166$ . The scaling rate is simply (x-1), and now the values actually scale at 8% per degree. In this example with a+2 degree anomaly, that would be 1.08 per degree or 1.08\*1.08, for 2 degrees of warming, which is equal to 1.166.

Equation 7 divides precipitation by its mean for each station-month, but many of these means are very small or zero, inflating anomalies and adding noise. Although this is mentioned briefly, its effect isn't explored or corrected.

The reviewer is correct that this can lead to larger values. To deal with this, we did not use stations which had a mean extreme precipitation amount less than or equal to 0.1 mm. So, we are always dividing by a mean that is greater than 0.1 mm. After normalizing the precipitation data, we did not find the presence of any abnormally large or unrealistic values. We will add a statement regarding this point in the revised paper.

Using monthly-mean dew point as a predictor helps correlation but weakens the physical link to rainfall extremes, which depend on short-term moisture and dynamics such as CAPE or large-scale ascent. Higher-frequency predictors would strengthen the physical interpretation.

Our aim to to provide a robust estimate of how extreme precipitation would be expected to change in a warming world. Of course, the more variables that one uses to predict extreme precipitation, the better those forecasts should be. However, we are not attempting to perform numerical weather prediction, but rather we want to provide a zoomed-out view of how something like 1 degree of further warming in a region would translate into expected average changes to

extreme precipitation for a given season. Further predictors and complexity can always be added, but that is beyond the scope of this paper.

The assumption that station-month maxima are statistically independent isn't fully demonstrated. Dependence across years or from climate modes like ENSO could still exist. Block-bootstrap or similar resampling methods would provide more realistic uncertainty estimates.

Thank you for this point. We will provide a better analysis of the statistical dependence of the data which we use in our modeling approach. However, we generate randomized predictions that exhibit the same temporal and spatial autocorrelation as the data itself. Therefore, we have already included the underlying data's statistical dependence in our model evaluation. We will include a more thorough analysis of the model performance in the revised paper.